# The histone demthylase KDM3A protects the myocardium from ischemia/reperfusion injury via promotion of ETS1 expression

Xin Guo[1,3], Bo-fang Zhang[1,3], Jing Zhang[2,3], Gen Liu[1], Qi Hu[1] & Jing Chen [1✉]

Our prior studies have characterized the participation of histone demethylase KDM3A in diabetic vascular remodeling, while its roles in myocardial ischemia/reperfusion (I/R) injury (MIRI) remain to be illustrated. Here we show that KDM3A was significantly downregulated in rat I/R and cellular hypoxia/reoxygenation (H/R) models. Subsequently, gain- and loss-of-function experiments were performed to investigate the effects of KDM3A in the settings of MIRI. KDM3A knockout exacerbated cardiac dysfunction and cardiomyocytes injury both in vivo and in vitro. The deteriorated mitochondrial apoptosis, reactive oxygen species, and inflammation were simultaneously observed. Conversely, KDM3A overexpression developed the ameliorated alternations in MIRI. Mechanistically, the MIRI-alleviating effects of KDM3A were associated with the enhancement of ETS1 expression. ChIP-PCR affirmed that KDM3A bound to the ETS1 promoter and removed dimethylation of histone H3 lysine 9 (H3K9me2), thus promoting ETS1 transcription. Our findings suggest that KDM3A is available for alleviating multi-etiologies of MIRI through the regulation of ETS1.

[1] Department of Cardiology, Renmin Hospital of Wuhan University; Cardiovascular Research Institute, Wuhan University; Hubei Key Laboratory of Cardiology, Wuhan 430000, China. [2] Department of Cardiology, The First College of Clinical Medical Science, China Three Gorges University & Yichang Central People's Hospital, Yichang 443003, China. [3] These authors contributed equally: Xin Guo, Bo-fang Zhang, Jing Zhang. ✉email: chenjing1982@whu.edu.cn

Epidemiological data from the World Health Organization demonstrate that approximately of 18.6 million patients who succumb to coronary vascular disease annually[1]. Timely revascularization, despite the benefits of limiting ischaemic injury and infarct expansion, governs the key endpoints that not only give rise to ischemia/reperfusion (I/R) episodes but also progress to chronic cardiac impairment[1–6]. Since achieving success in reperfusion to dampen ischemia, the focus of current experimental and clinical efforts has shifted towards the limitations of myocardial I/R injury (MIRI) to optimize the benefits of revascularization[3–5,7–10]. Although not completely clear, current investigations indicate that MIRI aetiologies consist of diverse cellular events with respect to apoptosis, reactive oxygen species (ROS), and inflammation[11–13]. Intriguingly, accumulating studies have illustrated that the dynamic regulation of epigenetic processes, such as histone methylation and acetylation, might be closely involved in these pathophysiologic progressions in response to MIRI[4,8,9,14]. Therefore, additional interference underlying epigenome-evoked multipathogenesis has the capacity to elicit promising targets and potential therapeutic avenues against MIRI.

Lysine specific demethylase 3 A (KDM3A, also called JMJD1A) is a well-known epigenetic activator functioning in target gene transcription via removal of the suppressive histone dimethylation mark of histone H3 at lysine 9 (H3K9me2) and is one member of the Jumonji C (JmjC) domain-containing dioxygenases (JMJC demethylases)[15–19]. In contrast to histone methylation[9,20], the modulation of histone demethylation with respect to MIRI occurrence is poorly investigated[4,21]. Our prior study confirmed that KDM3A-mediated H3K9me2 demethylation plays a crucial role in diabetic vascular remodeling[19]. The benefits derived from KDM3A inhibition on high insulin-induced vascular smooth cells malfunctions have also been shown, manifesting as a therapeutic approach due to pleiotropic potencies in the regulation of inflammation, apoptosis, and ROS[16]. However, little is known about whether and/or how KDM3A participates in cardiac damage, especially for MIRI. Recently, emerging studies have revealed that methylation is an important component of the epigenetic machinery in diabetic cardiomyopathy[22], myocardial infarction[23], and cardiac hypertrophy[24,25]. For instance, Mathias et al. suggested that KDM3A is significantly induced in failing myocardium, wherein its elevation potential but the inhibition negates the correlations with ANP and BNP transcription[26]. The potential mechanisms are ascribed at least in part to the constitutive alternation of histone methylation states on lysine residues (K), especially for the H3K9me2, involving multiple pathological events such as apoptosis and ROS generation[23]. As such the aberrant histone methylations implicated in KDM3A during MIRI have begun to gain attention based on the effects on H3K9me2. Given that H3K9me2 acts as a specific catalytic substrate of KDM3A, it is of interest to explore the underlying contributions of KDM3A in MIRI.

Erythroblastosis virus E26 oncogene homologue 1 (ETS1) has been well-recognized as a multifaceted mediator of inflammation, apoptosis, and ROS[27–31]. It has been reported that KDM3A-mediated transactivation of ETS1 occurs via a decrease in H3K9me2 is linked to cancer metastasis[18]. Specifically, Bian et al. demonstrated that mitochondrial apoptosis is markedly worsened in response to an absence of ETS1, increasing cardiac vulnerability to I/R injury[31]. However, the comprehensive roles of ETS1 in MIRI are far from clear, and more importantly, there is no evidence on whether KDM3A exerts cardioprotective effects against I/R in an ETS1-dependent manner. In the present study, by utilizing gain- and loss-of-functional approaches, we identified KDM3A as a novel protective regulator against MIRI both in vivo and in vitro. The mechanism underlay epigenetic modification of ETS1 via repression of histone methylation marker H3K9me2 on the promoter, thereby directly involving in the cell-fate outcome like survival, mitochondria-relevant apoptosis, ROS and inflammation. Thus, KDM3A emerges as a potent therapeutic candidate for the prevention of MIRI.

## Results

**KDM3A expression is downregulated after MIRI**. To unravel whether KDM3A is involved in MIRI, we first measured KDM3A expression in I/R-treated rat hearts and H/R-exposed NRCMs. We observed that the expression of KDM3A at the protein (Fig. 1a) and mRNA (Fig. 1b) levels was progressively downregulated in rat hearts 24 h after reperfusion compared to the sham-operated group. Accordingly, double IF for KDM3A (green) and α-actinin (red) further revealed decreased expression and cytoplasmic/nuclear location of KDM3A in I/R hearts relative to sham hearts (Fig. 1c). A similar reduction in KDM3A was observed in isolated NRCMs subjected to an in vitro model of H/R (Fig. 1 d–f). This evidence implicates the possible contributions of KDM3A during MIRI.

**KDM3A overexpression protects the myocardium from I/R injury**. To examine the potential functions of KDM3A in MIRI, intramyocardial infection with AdKDM3A or AdGFP was performed before establishing the MIRI model. The hallmarks of MIRI were assessed by functional and morphological alterations. As shown in Fig. 2a, the protein levels of KDM3A were markedly increased after AdKDM3A delivery. Then, myocardial infarct size as visualized by Evans blue + TTC staining and that represented by the ratio of IA to LV was profoundly attenuated in AdKDM3A-infected rats after I/R injury as compared to the I/R group (Fig. 2b). Cardiac function as assessed by echo assay at post-I/R suggested an obvious enhancement in AdKMD3A-transduced rats, as determined by elevated LVEF% (Fig. 2c) and LVFS% (Fig. 2d) compared to the I/R group. In consistent with these phenomena, the lower activities of LDH (Fig. 2e) and CK-MB (Fig. 2f) resulted in less severe necrosis in KDM3A-overexpressing myocardium than in I/R rats. Notably, AdGFP transfection exerted no apparent effects on the above parameters of MIRI (vs. the I/R group, $p > 0.05$). These findings confirm that KDM3A overexpression ameliorates I/R-induced cardiac dysfunction, infarct size enlargement, and myocardial necrosis in vivo.

**KDM3A overexpression attenuates apoptosis, ROS, and inflammation in MIRI**. Apoptosis originating from mitochondrial malfunction is widely regarded as a potent indicator of MIRI, accompanied by disruption of mitochondrial structure/function and increased levels of cytochrome $c$ release, Bax/Bcl-2 ratio, and cleaved caspase-9/3[7,32]. To determine the effect of KDM3A on mitochondria-associated apoptotic cascades, TUNEL staining for apoptotic cells in parallel with TEM detection and ATP content assay for mitochondrial structure and function were measured respectively after MIRI. Moreover, we assessed the protein expressions of cytoplasmic/mitochondrial cytochrome $c$, Bax, Bcl-2, cleaved caspase-9, and cleaved caspase-3. As shown in Fig. 3a, AdKDM3A infection in I/R myocardium resulted in fewer TUNEL-positive cells (Fig. 3a, b), less swelling, fewer mitochondrial vacuoles (Fig. 3c) and a marked increase in ATP content (Fig. 3d) relative to those of the I/R group. Similarly, diminished levels of mitochondrial apoptosis indicators, as displayed by reductions in cytoplasmic cytochrome $c$, Bax, and cleaved caspase-9/3 and increased mitochondrial cytochrome $c$ and Bcl-2, were substantially observed after AdKDM3A delivery in MIRI (Fig. 3e, f; vs. I/R group, $p < 0.05$).

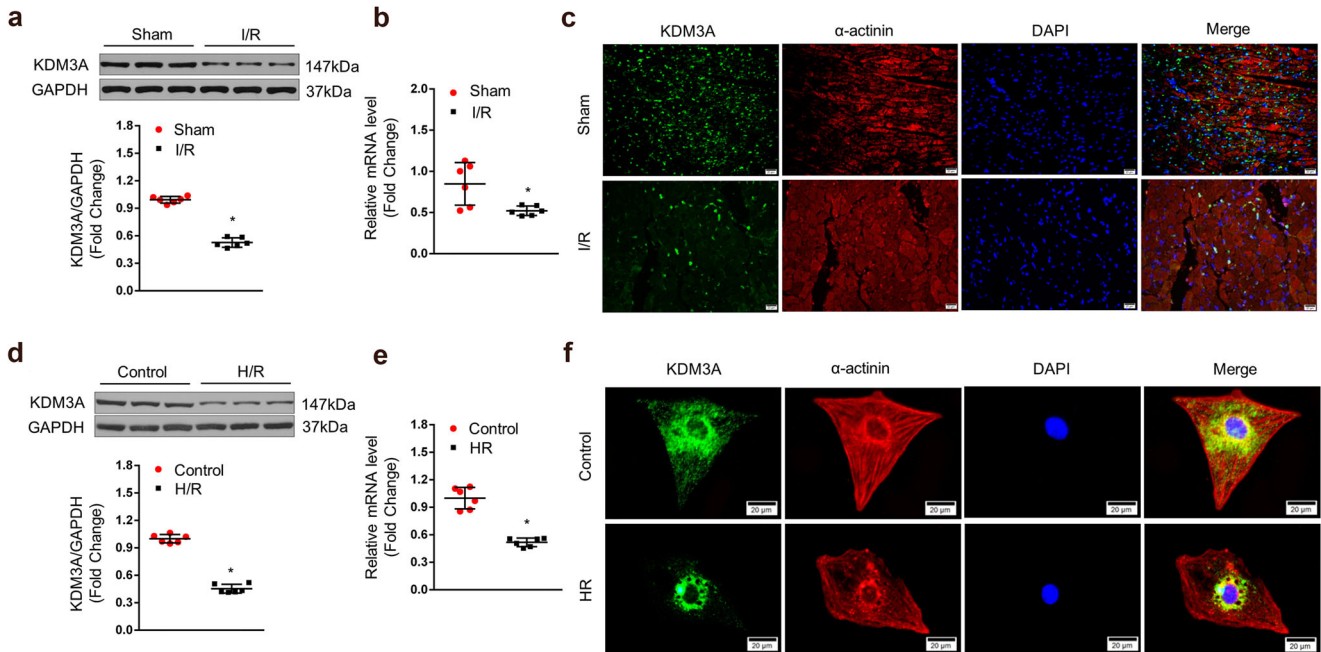

**Fig. 1 KDM3A expression was downregulated after MIRI. a, b** Levels of KDM3A protein (detected by Western blot assay) (**a** top panels are representative blots; bottom bar graphs show quantitative results) and mRNA (detected by qRT-PCR) (**b**) in rat heart at post I/R injury or a sham-operation (n = 6, *P < 0.05 vs. sham). **c** Representative dual immunofluorescence staining of KDM3A (green) and α-actinin (red) in heart sections with or without I/R injury (n = 4). The nuclei were stained with DAPI (blue). The upper panel shows heart sections from sham-operated animals; the lower panel shows heart sections from the I/R group. Scale bar = 20 μm. **d, e** Levels of KDM3A protein (**d** top panels are representative blots; bottom bar graphs display quantitative results) and mRNA (**e**) (n = 6, *P < 0.05 vs. control) in cultured NRCMs in the absence or presence of H/R injury. **f** Double immunofluorescence images of KDM3A (green) and α-actinin (red) in NRCMs treated with H/R or normoxic controls (n = 4). The top panels and bottom panels display cardiomyocytes subjected to normoxic control and H/R injury, respectively. Scale bar = 20 μm.

ROS generation and inflammation also act as preferential agents in response to myocardial I/R insult despite apoptotic cascades[5,33]. To further reveal KDM3A's participation in modulating ROS formation, in situ DHE staining together with the activity of antioxidative SOD and the content of the oxidative marker MDA were determined essentially. As shown in Fig. 3g, DHE staining revealed debilitated red fluorescence and a considerable decrease in $O^{2-}$ generation in AdKDM3A-overexpressing rats relative to the I/R controls 24 h after reperfusion. In parallel, limited ROS activity as manifested by decreased MDA and augmented SOD was also verified (Fig. 3h, i, AdKDM3A + I/R vs. I/R group, p < 0.05). Furthermore, concentrations of the well-known proinflammatory mediators (IL-6 and TNF-α) remained dramatically lower levels in the AdKDM3A + I/R group than in the I/R group (Fig. 3j, k, p < 0.05). Notably, no significant differences were observed in apoptosis-/ROS-/inflammation-associated parameters between the I/R and the AdGFP +I/R groups (p > 0.05). Thus, the above gain-of-function evidence implies that KDM3A upregulation serves as a pivotal agent in suppressing apoptosis, ROS, and inflammation, in which it confers multiple protective effects against MIRI.

**KDM3A knockout exacerbates MIRI in vivo.** To further address the functions of endogenous KDM3A in MIRI in vivo, a novel $KDM3A^{-/-}$ rat strain was generated using the well-identified CRISPR/Cas9 technology (Supplementary Fig. 1a–d). Western blotting was performed to validate the absence of endogenous KDM3A protein in $KDM3A-/-$ rat hearts (Fig. 4a). Subsequently, the $KDM3A^{-/-}$ and WT littermates ($KDM3A^{+/+}$) were subjected to an I/R procedure or sham operation. Interestingly, $KDM3A^{-/-}$ hearts exhibited drastic increases in myocardial infarct size (Fig. 4b), cardiac malfunction (Fig. 4c, d) and necrosis

(Fig. 4e) compared to $KDM3A^{+/+}$ rats after reperfusion. The impacts of KDM3A deficiency on apoptosis, mitochondrial structure, ROS and inflammation on MIRI were also examined. The elevated number of TUNEL-positive cells (Fig. 5a, b), worsened mitochondrial structure (Fig. 5c), reduced ATP content (Fig. 5d) and promoted mitochondria-triggered apoptotic cascades (Fig. 5e, f), which were more pronounced in $KDM3A^{-/-}$ rats than that in $KDM3A^{+/+}$ animals. Similarly, ROS generation, as indicated by in situ DHE staining, SOD activity and MDA content, was dramatically augmented in $KDM3A^{-/-}$ rat hearts compared to $KDM3A^{+/+}$ rat hearts (Fig. 5g–i). Furthermore, $KDM3A^{-/-}$ rats exhibited much higher serum levels of IL-6 and TNF-α than $KDM3A^{+/+}$ rats under I/R conditions (Fig. 5j, k). No significant differences in those parameters were observed between sham-operated $KDM3A^{+/+}$ and $KDM3A^{-/-}$ rats. Therefore, in contrast to the consequences of KDM3A upregulation in vivo, loss of KDM3A induces the rat heart to be more susceptible to I/R injury, the mechanisms of which are ascribed to the enhancement of mitochondrial apoptosis, ROS, and inflammation.

**The effects of KDM3A on H/R-exposed cardiomyocytes in vitro.** Given that KDM3A deletion worsens while KDM3A overexpression ameliorates MIRI in vivo, we subsequently performed controlled gain- and loss-of-function experiments to define the potencies of KDM3A in H/R-exposed cultured NRCMs in vitro subsequently. We infected the cells with either AdKDM3A to overexpress KDM3A or AdshKDM3A to knockdown KDM3A. AdGFP and AdshRNA transduction served as the negative controls (supplementary Figs. 1, 2). Next, these cells were subjected to the H/R model or a normoxic control. As expected, H/R exposure resulted in a strong decrease in cell viability (Fig. 6a, b) and augmentations of LDH/CK-MB activities (Fig. 6c, d) compared to the

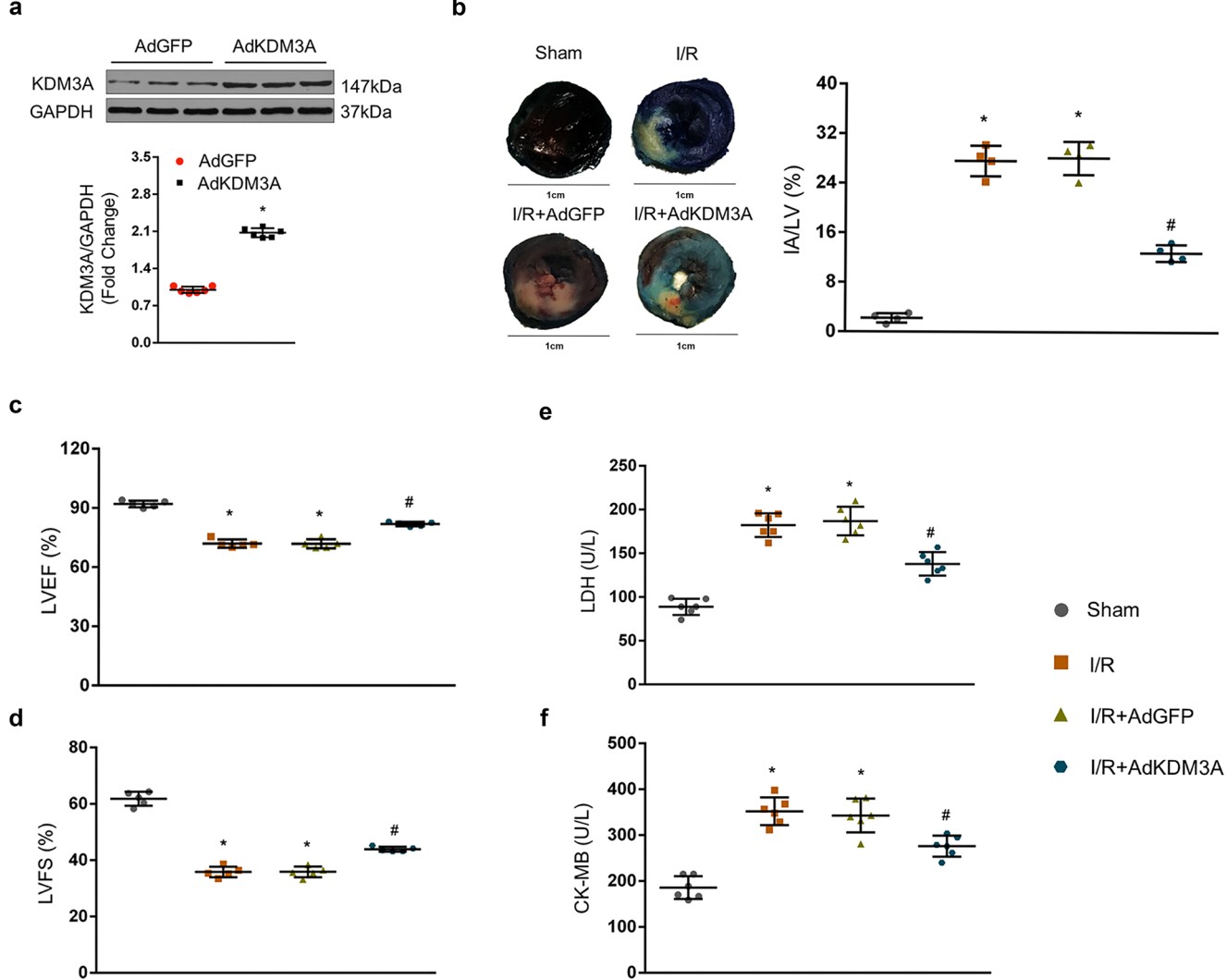

**Fig. 2 KDM3A overexpression protects the myocardium against I/R injury. a** Protein levels of KDM3A in rats intramyocardially transduced with AdKDM3A or AdGFP (n = 6, *P < 0.05 vs. AdGFP). Top: representative blots; bottom: quantitative results. **b** Representative section images (left panel) and averaged data of the ratio of IA to total left ventricular area (right panel) (detected by Evans blue+TTC staining) (n = 4, *P < 0.05 vs. I/R). The Evans blue-stained area (dark blue) represents the nonischemia/reperfusion area; TTC/Evans blue-negative regions (pale white) represent the infarcted myocardium. Scale bar = 1 cm. **c, d** Echocardiographic results of left ventricle ejection fraction (LVEF) and fractional shortening (LVFS) indicated cardiac function after KDM3A upregulation in response to MIRI or sham operation (n = 5, *P < 0.05 vs. sham; #P < 0.05 vs. I/R or I/R + AdGFP). **e, f** Myocardial death was estimated by the serum LDH/CK-MB concentrations with or without MIRI (n = 6, *P < 0.05 vs. sham; #P < 0.05 vs. I/R or I/R + AdGFP).

controls. However, these outcomes were markedly ameliorated after AdKDM3A infection but were further exacerbated in KDM3A-knockout cardiomyocytes (Fig. 6a–d). Next, we examined the effects of controlled KDM3A on mitochondria-medicated apoptosis, ROS production, and inflammation in response to H/R injury. Similar to the in vivo evidence, H/R-induced mitochondrial apoptosis and apoptotic cascade exhibited a marked reduction after AdKDM3A infection but induced the opposite effect in KDM3A-knockdown cells, as indicated by flow cytometry for the apoptotic rate (Fig. 6e), ATP content (Fig. 6f) and the protein detection of cytoplasmic/mitochondrial cytochrome *c*, Bax, Bcl-2 and cleaved caspase-9/3 (Fig. 6g–j). Consistently, KDM3A overexpression via AdKDM3A transfection revealed significantly diminished ROS relative to the controls after H/R, as indicated by the mean fluorescence intensity (MFI) of DHE staining (Fig. 6k, l) and alterations in MDA concentration (Fig. 6m) and SOD activity (Fig. 6n). In contrast, AdshKDM3A transfection remarkably exacerbated ROS production (Fig. 6k–n). Similarly, the concentrations of IL-6 and TNF-α were much higher following KDM3A knockdown than in the controls,

but were significantly rescued after KDM3A overexpression in response to the H/R injury (Fig. 6o, p). In combination with the data from both in vivo and in vitro experiments, KDM3A exerts beneficial effects in MIRI, particularly involving the limitations of apoptosis, ROS, and inflammation.

**KDM3A is involved in MIRI by regulating the expression of ETS1 both in vivo and in vitro.** The above results suggest that KDM3A confers favorable potency against MIRI, but the potential molecular mechanisms whereby this occurs remain unknown. To address this issue, we examined the transcriptional factor ETS1, as a known target of KDM3A and an important mediator implicated in apoptosis, ROS, and inflammation[18,27,28,31]. qRT-PCR and western blotting revealed that I/R dramatically reduced the mRNA and protein expression of ETS1, which were markedly increased after AdKDM3A transfection compared to the I/R group in vivo (Fig. 7a, b). However, in the loss-of-function data, the I/R-induced decrease in ETS1 at both the mRNA and protein levels was much lower in *KDM3A*[−/−] rats than that in *KDM3A*[+/+] controls

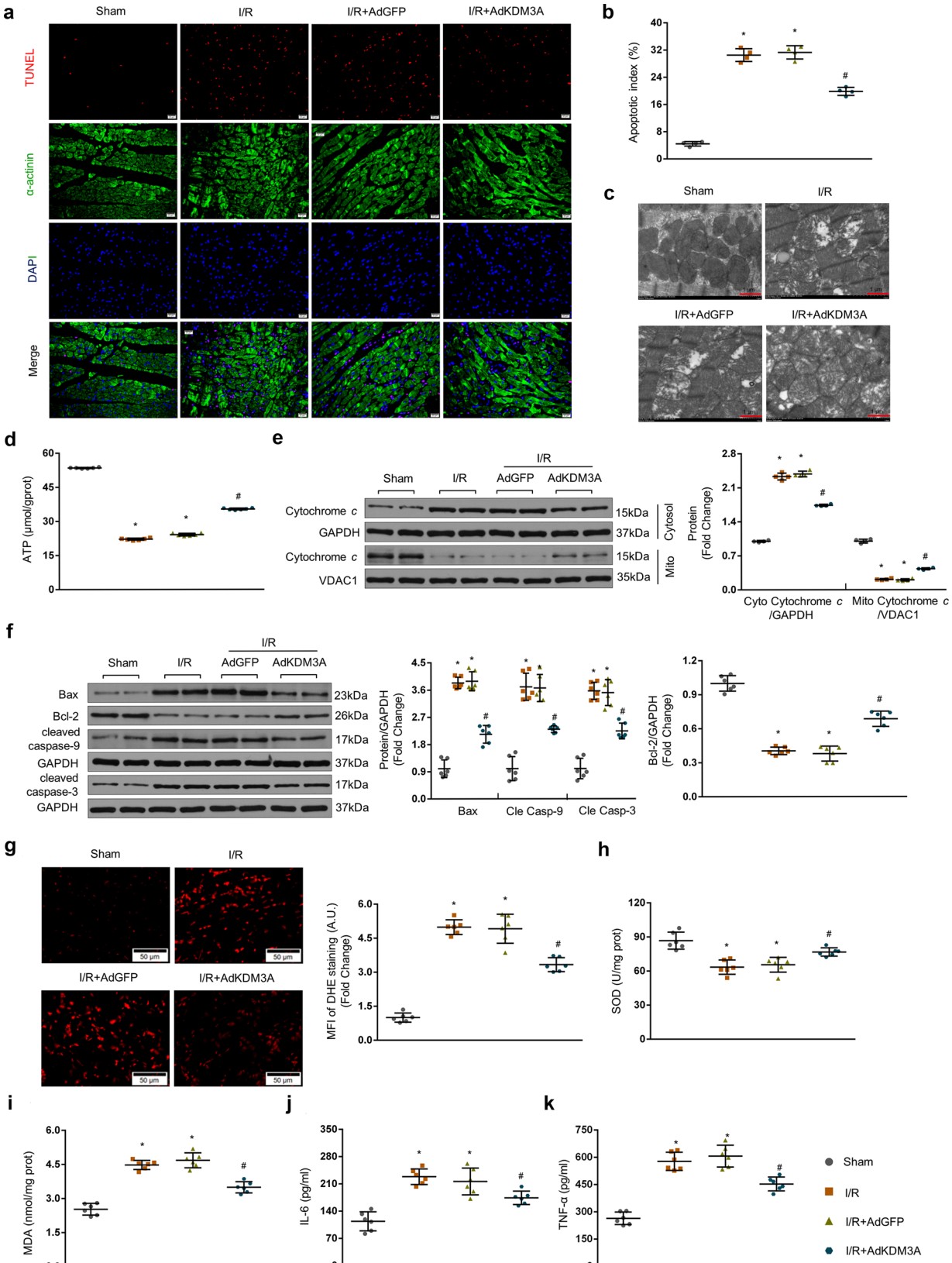

(Fig. 7c, d). To further confirm these in vivo results, the NRCMs were subjected to either AdKDM3A or AdshKDM3A, followed by H/R. Consistent with the in vivo findings, KDM3A knockdown potentiated H/R-induced repression of ETS1 transcription, whereas this elimination was rescued by KDM3A overexpression (Fig. 7e–h). Furthermore, in accordance with the consequences of

KDM3A on ETS1 expression, H3K9me2, a repressive histone methylation marker and a specific substrate of KDM3A-mediated catalysis, was also regulated by KDM3A in the opposite direction under MIRI both in vivo and in vitro (Fig. 7a–h).

Following experiments performed by ChIP-PCR assay were used to further determine whether ETS1 activation by KDM3A was due to

**Fig. 3 KDM3A overexpression attenuates apoptosis, ROS and inflammation in MIRI. a**, **b** Representative images (**a**) and averaged data of the percentage of TUNEL-positive cells (**b**) in rat hearts with or without viral transduction subjected to I/R injury ($n = 4$). TUNEL staining (green) indicates apoptotic nuclei; DAPI counterstaining (blue) indicates total nuclei. Scale bar = 20 μm. The apoptotic index (AI) is presented as the percentage of TUNEL-positive nuclei to the total number of nuclei. **c** Representative mitochondrial morphologies detected by TEM exhibiting the divergent status of swollen mitochondria, disrupted mitochondrial cristae and disorganized myofibrils with or without viral transduction either subjected to I/R injury or not ($n = 3$). Scale bar = 1 μm. **d** Mitochondrial function determined by myocardial ATP production ($n = 6$). **e**, **f** Cardiac protein levels of mitochondria-associated apoptotic mediators, including Bax ($n = 6$), Bcl-2 ($n = 6$), cleaved caspase-9/3 ($n = 6$), cytoplasmic cytochrome $c$ ($n = 4$) and mitochondrial cytochrome $c$ ($n = 4$), detected by western blot assay. Left panels show representative blots; right bar graphs show quantitative results ($n = 6$). **g** Representative images (left panel) and mean fluorescence intensity (MFI) (right panel) of DHE staining on frozen myocardial tissue sections, indicating ROS production ($O_2^-$ in particular) in the presence or absence of MIRI ($n = 6$). Scale bar = 50 μm. **h**, **i** MDA levels and SOD activity in cardiac tissue ($n = 6$). **j**, **k** Serum levels of proinflammatory mediators (IL-6 and TNF-α) were detected by commercial ELISA kits in the presence or absence of MIRI with or without viral transduction ($n = 6$). *$P < 0.05$ vs. sham; #$P < 0.05$ vs. I/R or I/R + AdGFP.

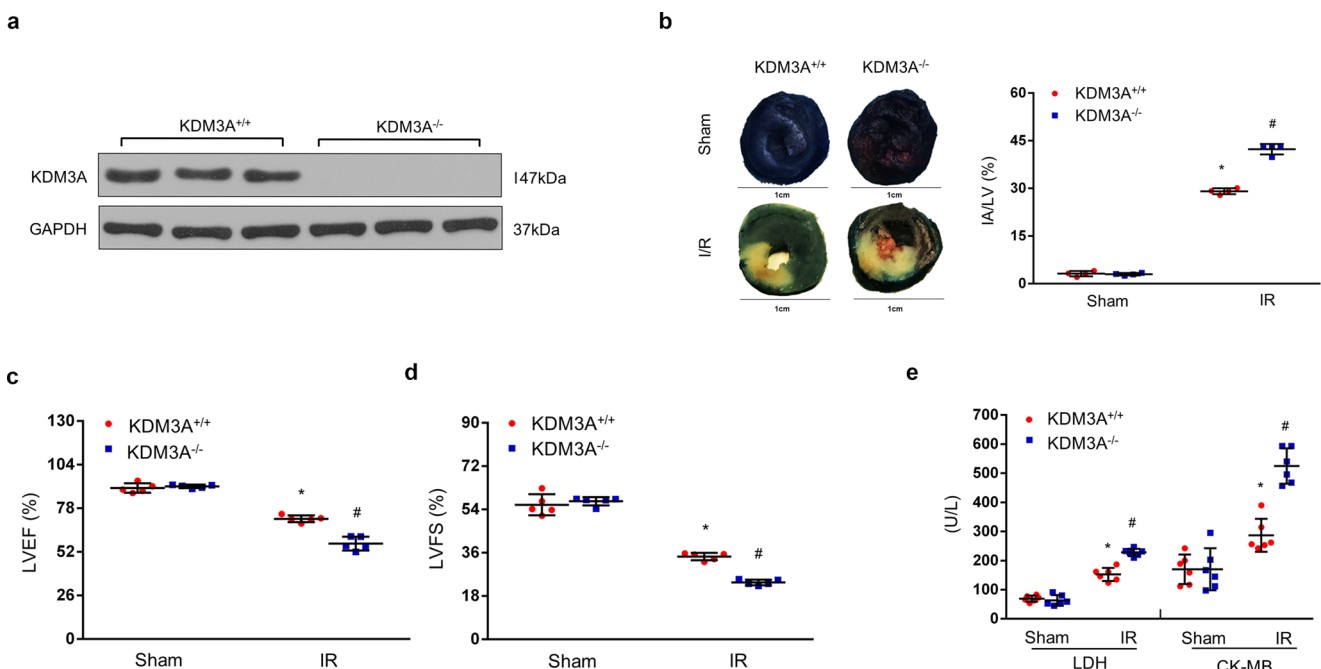

**Fig. 4 KDM3A knockout exacerbates I/R-induced myocardial injury. a** Representative western blots of KDM3A protein expression in heart tissue from KDM3A$^{+/+}$ and KDM3A$^{-/-}$ ($n = 6$). **b** Representative section images (left panel) and averaged data of the ratio of IA to total left ventricular area (right panel) after sham or I/R surgery (n = 4). Scale bar = 1 cm. **c**, **d** Statistical analysis of echocardiographic parameters (LVEF and LVFS) indicating left ventricular function upon MIRI or sham operation in the indicated groups ($n = 5$). **e** Myocardial damage was estimated by LDH/CK-MB concentrations in the serum with or without MIRI ($n = 6$). *$p < 0.05$ vs. KDM3A$^{-/-}$/sham; #$p < 0.05$ vs. KDM3A$^{+/+}$/IR.

the epigenetic reduction in H3K9me2 levels. As shown in Fig. 8a–d, ChIP-PCR demonstrated that the H3K9me2 levels were increased on the ETS1 gene proximal promoter in H/R-exposed cardiomyocytes compared to controls. Parallel suppression of KDM3A occupancy on the ETS1 promoter was also observed following H/R injury. In addition, KDM3A overexpression greatly repressed the H/R-induced increase in H3K9me2 levels while augmenting KDM3A recruitment to the ETS1 promoter (Fig. 8a–d); in contrast, there was a simultaneous elevation of H3K9me2 and a reduction in KDM3A on the ETS1 promoter after KDM3A knockdown in the H/R-treated myocardium (Fig. 8a–d). Meanwhile, neither H3K9me2 nor KDM3A was markedly changed with respect to the GAPDH promoter, which was used as a negative control (Fig. 8e–h). Combined, these data indicate that H3K9me2 accumulation on the ETS1 promoter, closely mediated by KDM3A, is responsible for ETS1 transactivation in the myocardium following the H/R paradigm.

**KDM3A ameliorates MIRI in an ETS1-dependent manner.** Finally, we examined whether KDM3A exertes multiple cardioprotective effects through an ETS1-dependent pathway. We constructed

three kinds of siRNA-ETS1, and siETS1-02 exerted the strongest inhibitory on ETS1 protein expression, as measured by western blotting assay (Fig. 9a). Next, siETS1-02 was used in AdKDM3A-infected cardiomyocytes. Twenty-four hours after reoxygenation, siETS1 treatment greatly reversed anti-H/R effects of KDM3A, as shown by the following observations: (i) reduced cell viability(-Fig. 9b); (ii) increased levels of LDH and CK-MB activities(Fig. 9c); (iii) elevated MFI of DHE staining (Fig. 9d, e), increased apoptotic rate (Fig. 9f, g) and potentiated IL-6/TNF-α releases (Fig. 9h, i) in ETS1-administered KDM3A-overexpressing NRCMs compared to the AdKDM3A-infecting controls during H/R. Taken together, our data indicate that KDM3A elicits protective contributions in apoptosis-, ROS- and inflammation-associated processes, largely dependent on the activation of ETS1 during MIRI.

**Discussion**
Apoptosis-, ROS- and inflammation-induced myocardial loss represents the most pivotal agents in MIRI[2,7,9,10,14,34–36]. In this way, acute impairment is dramatically induced and allows for subsequent cardiac remodelling post I/R[2,3]. In terms of KDM3A

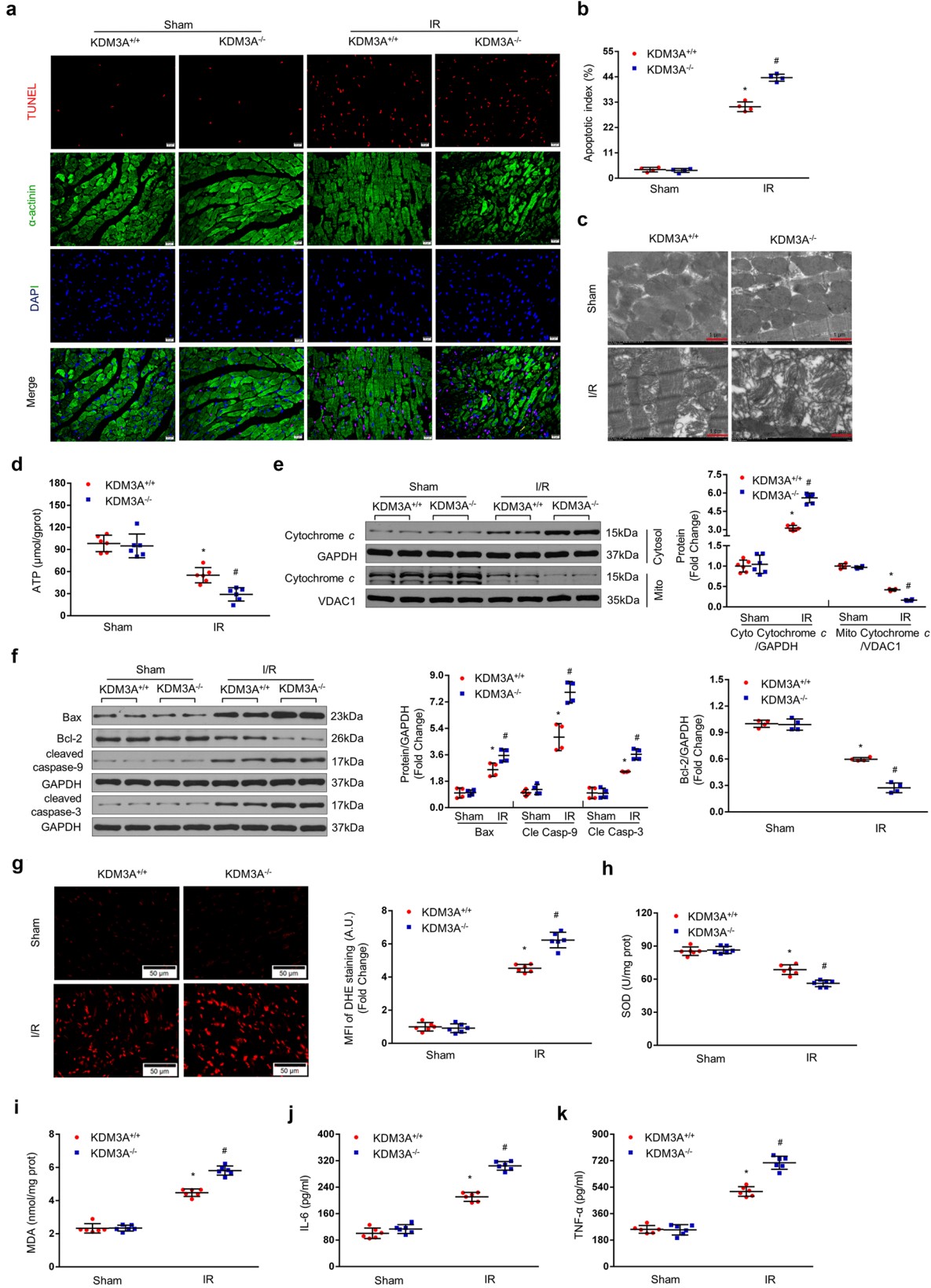

in modulating various pathological processes, such as vascular remodeling and ischemic stress[15,16,19], we speculated that KDM3A may function as a decisive mediator in the pathogenesis of MIRI. The purposes of our present study focus on (1) whether KDM3A was involved in MIRI; if so, (2) how it impacts multiple pathogeneses of MIRI, yielding protective effects and (3) to uncover the underlying mechanism, particularly ETS1, anchoring the I/R-inhibitory roles of KDM3A.

AMI is one of the leading causes of morbidity and mortality worldwide[2,7,9,34,36]. Diverse pharmacological and surgical therapies could efficiently reverse ischemia, whereas ineluctable I/R damage paradoxically counteracts reperfusion advances and

**Fig. 5 KDM3A knockout exacerbates apoptosis, ROS and inflammation in MIRI. a** Myocardial apoptosis determined by TUNEL staining. TUNEL staining (red) revealing apoptotic nuclei. DAPI counterstaining (blue) visualized total nuclei. Scale bar = 20 μm. **b** The apoptotic index (AI) was calculated using Image-Pro Plus software ($n = 4$). **c, d** Representative mitochondrial morphologies detected by TEM ($n = 3$, Scale bar = 1 μm) and mitochondrial function measured by ATP production ($n = 6$) in the presence or absence of MIRI in the indicated groups. **e** ($n = 6$) and **f** ($n = 4$), Cardiac protein levels of mitochondria-associated apoptotic mediators by western blot assay. Left panels show representative blots; right bar graphs show quantitative results. **g** Representative images (left panel) and mean fluorescence intensity (MFI) (right panel) of DHE staining on frozen myocardial tissue sections. $n = 6$. Scale bar = 50 μm. **h, i** Levels of pro-oxidative MDA and antioxidative SOD ($n = 6$). **j, k** The release of proinflammatory mediators (IL-6 and TNF-α) determined by ELISA kits (n = 6). *$p < 0.05$ vs. KDM3A $^{-/-}$/sham; #$p < 0.05$ vs. KDM3A $^{+/+}$/IR.

strikingly facilitates deleterious cardiac performance[2,7,9,34,36]. Although therapeutic methods for minimizing MIRI are currently finite, emerging evidence indicates that the vigorous stimulation of apoptotic cascades, secondary to incipient mitochondrial malfunction, induces critical mechanisms that aggravate myocardial loss in I/R settings[35]. Among these factors, the evoked Bax/Bcl-2 ratio cooperatively results in the formation of mitochondrial pores, parallel to the aggravated cytochrome $c$ release and an irreversible apoptotic cascade[10,35]. Moreover, ROS accumulation originating from mitochondrial impairments can also subsequently amplifies cytotoxic outcomes[10]. Once oxidative stress disrupts the balance between antioxidant defence and ROS in favour of the latter, it constitutes a crucial aetiological component of MIRI[10]. Excessive ROS elicits an imbalanced redox status, particularly as reflected by the inactivation of the antioxidant enzyme SOD and increased MDA levels[11,14,33]. Furthermore, inflammation has been proposed to aggravate MIRI[5]. Consistently, our data revealed that I/R leads to disturbed mitochondrial structure and function in concert with elevated inflammation, ROS, and the mitochondria-associated apoptotic cascade. Interestingly, KDM3A overexpression ameliorated these abnormalities, while KDM3A knockout exacerbated these abnormalities both in vivo and in vitro. Therefore, we believe that KDM3A protects cardiomyocytes from I/R injury particularly by interfering with ROS, apoptosis, and inflammation. Interventions involved in multiple pathogenesis, more specifically for KDM3A, may glean unrecognized modalities and provide impetus in mitigating MIRI.

Epigenetic modulations that occur within MIRI principally involve DNA methylation, noncoding RNAs and posttranslational modifications, such as histone methylation[8–10,14,20]. Specific alternations of histones affect the chromatin architecture switch between opening and condensation, regulating target gene activation or repression beyond genomics[8,9,19,20,23,25]. Among these, histone H3 is one of the four categories of core histones. The activities of histone demethylases (HDMs), including LSD1 and the JMJC domain-containing proteins[15,19,37], and histone methyltransferases (HMTs), including SUV39H1, G9a, and others[8,9,23,38] remove or deposit methyl groups at lysine 9 of H3 (H3K9) from pathogenic gene promoters respectively, that are strictly relevant to facilitating or repressing target genes accordingly. In contrast to the well-identified contributions of HDMs and HMTs in diabetes and cardiac hypertrophy[22,24,25,39], little is known regarding their properties in MIRI. However, the discovery of HMTs, i.e., SUV39H1, G9a, Smyd2, and others, has substantially shifted this paradigm. For instance, a promising observation based on the selective knockout of caveolin-1 was verified in a prior study regarding the capacity of worsened outcome after IPC during MIRI by stimulating G9a[40]. Erlinge et al. recently provided direct clues that IPC within MIRI confers epigenetic inhibition of *Mtor* and causal enhancement of autophagy through G9a-dependent H3K9me2 augmentation[20]. A similar finding underlying the ROS- and apoptosis-suppressed actions as a result of SUV39H1 silencing or inhibition is observed both in MIRI and AMI, and potential mechanisms include the

SIRT1 trans-repression by erasing H3K9me3 from its promoter and the subsequent normalization of SIRT1 in cardiomyocytes[9,23]. In terms of the opposite biochemical action of HDMs relative to HMTs, it is tempting to speculate the possible roles of HDMs in response to MIRI, although they are not entirely clear. In the present study, we provided evidence for the first time that KDM3A, a member of the JMJC domain-containing proteins that belong to the HDMs family, was sufficient to recapitulate multiple protective potencies by removing the repressive H3K9me2 histone mark. This study suggests that histone demethylation seems to be another crucial pathogenetic mechanism irrespective of histone transmethylation during MIRI. Another critical goal of future research is to mechanistically identify direct or indirect sensors in response to KDM3A that regulate KDM3A to convey pleiotropic protection against MIRI.

To clarify the mechanisms underpinning the I/R-reduced roles of KDM3A, ETS1 drew our great attention. ETS1 is a member of the ETS family featuring a large group of transcription factors. In a previous study, KDM3A-mediated transactivation of ETS1 was shown to occur via the H3K9me2 decrease in the genomic region flanking the promoter and was especially linked to cancer development. Notably, many studies to date have verified ETS1 as a multifunctional mediator implicated in vascular remodeling[28], renal injury in salt-sensitive hypertension[41] and I/R disorder[31]. For instance, ETS1 stimulation is responsible for apoptotic limitation in renal I/R[29]. Diminished ROS generation and inflammation in Ang II-induced vascular remodeling were observed in ETS1-deficient mice[30]. On the other hand, miRNAs, which are epigenetic mediators, have been suggested to regulate ETS1 in various of ways. Gareri et al. demonstrated that miR-125a-5p modulates the phenotypic switch of vascular smooth cells by directly targeting ETS1[42], and other miRNAs, such as miR-221, appear to modify ox-LDL-induced endothelial apoptosis by affecting ETS1[43]. Nevertheless, the involvement of ETS1 in cardiac I/R has received less attention. A recent report demonstrated that a lack of ETS1 provides the possibility of reducing mitochondrial apoptosis under MIRI, as significantly characterized by alterations in of Bax, Bcl-2, and caspase-3[31]. Because of the compelling roles of ETS1 in MIRI and the potential of ETS1 to be a KDM3A-targeted gene, and the versatile activities of ETS1 in inflammation, ROS, and apoptosis, we assessed possible correlations between ETS1 and KDM3A in MIRI. As expected, our study found that KDM3A directly targeted ETS1 by binding to its promoter region, in concert with the abrasion of H3K9me2. More intriguingly, KDM3A overexpression resulted in higher ETS1 transcription in concomitant with reduced apoptosis, ROS, and inflammation, while KDM3A knockdown reversed these alterations. In addition, by applying ETS1-siRNA to the H/R-exposed cardiomyocytes, KDM3A overexpression blunted the protective outcomes and restored apoptosis, ROS, and inflammation. Therefore, our findings reinforce that the protective roles of KDM3A against MIRI are, at least partially, ascribed to the epigenetic transactivation of ETS1 and that the exact participation of ETS1 in MIRI is constitutively expanded, in addition to the known apoptosis-limited effect. More studies are still warranted

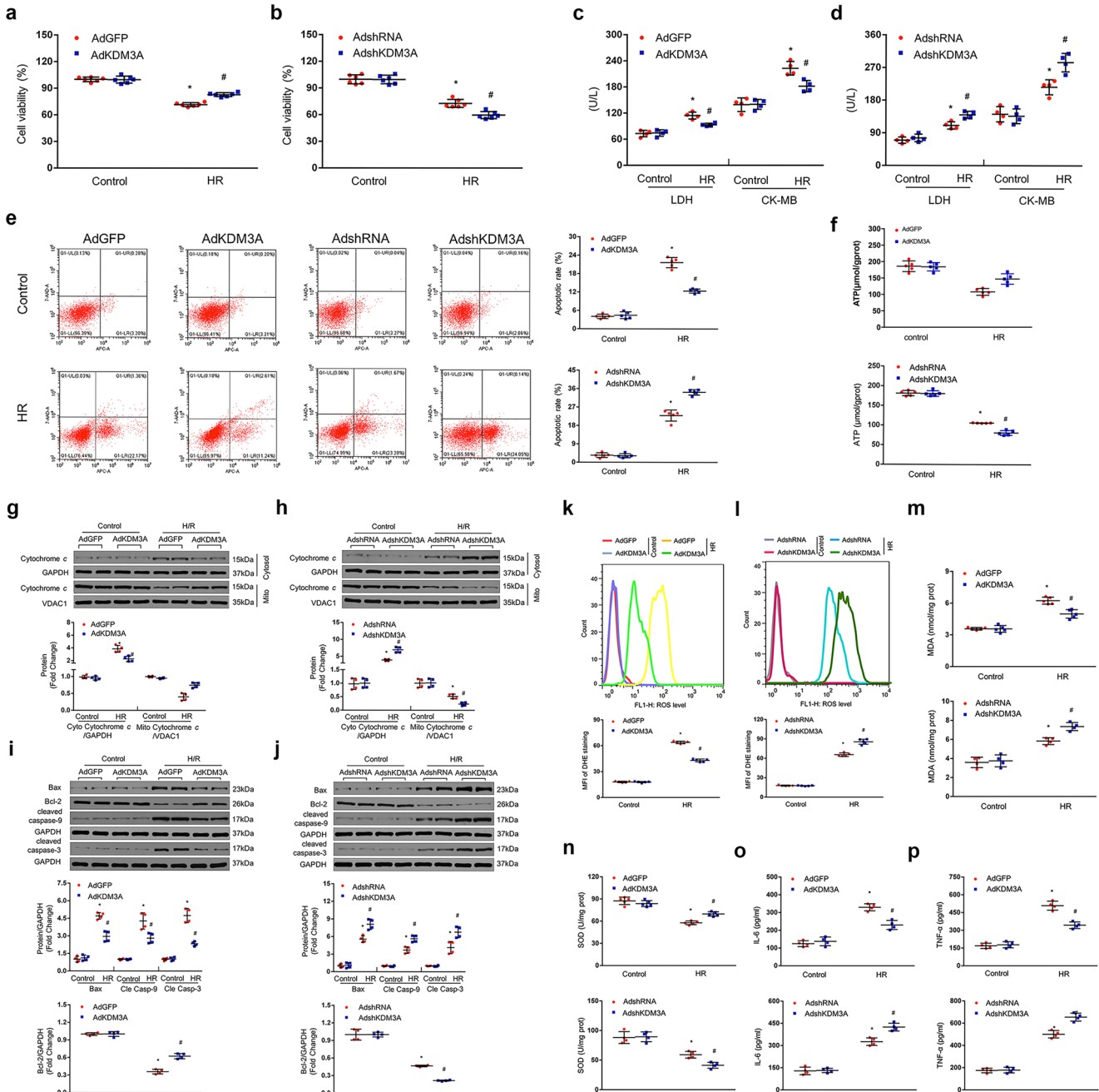

**Fig. 6 The effects of KDM3A on H/R-exposed cardiomyocytes in vitro.** Cell viability analysed by MTT (**a**, **b**) and culture medium LDH/CK-MB concentration (**c**, **d**). $n = 6$ per group, $*P < 0.05$ vs. AdGFP or AdshRNA/Control; $\#P < 0.05$ vs. AdGFP or AdshRNA/H/R. **e** Representative images (upper panel) and quantitative results (lower panel) of apoptotic cardiomyocytes (detected by flow cytometry assay) infected with AdKDM3A or AdshKDM3A in response to H/R or control ($n = 5$). **f** ATP production in NRCMs infected with AdKDM3A or AdshKDM3A either exposed to H/R or not ($n = 5$). **g–j** Western blotting assay showing the levels of cytoplasmic cytochrome *c*, mitochondrial cytochrome *c* (**g**, **h**) Bax, Bcl-2 and cleaved caspase-9/3 (**i**, **j**) in primary cardiomyocytes infected with AdKDM3A or AdshKDM3A with or without H/R injury ($n = 4$). Upper panel: representative bots; lower panel: quantitative analysis. $*P < 0.05$ vs. AdGFP or AdshRNA/Control; $\#P < 0.05$ vs. AdGFP or AdshRNA/H/R. **k**, **l** ROS accumulation detected by DHE staining and flow cytometry (upper panel) and quantified MFI (lower panel) in NRCMs ($n = 5$). **m**, **n** Levels of pro-oxidative MDA and antioxidative SOD in NRCMs ($n = 4$-5). **o**, **p** Quantification of proinflammatory cytokines (IL-6 and TNF-α) in AdKDM3A-/AdshKDM3A-infected cardiomyocytes either exposed to H/R or not ($n = 4$). $*P < 0.05$ vs. AdGFP or AdshRNA/Control; $\#P < 0.05$ vs. AdGFP or AdshRNA/H/R.

to explore whether there are other direct targets of KDM3A irrespective of the ETS1 in MIRI.

The pleiotropic activities of KDM3A in the regulation of migration, proliferation, apoptosis, inflammation, and ROS have been elucidated in our previous studies[16,19]. For instance, KDM3A inhibition is considered a promising avenue for the

restoration of vascular function in diabetes-mediated metabolic memory. In this study, the Rho/ROCK2 and AngII/AGTR1/MAPKs (ERK1/2 and JNK1/2, but not p38MAPK) signaling pathways were identified as two of the KDM3A-mediated effectors, and AGTR1/ROCK2 acts as a KDM3A-targeted gene by controlling H3K9me2 in their proximal promoters. In addition,

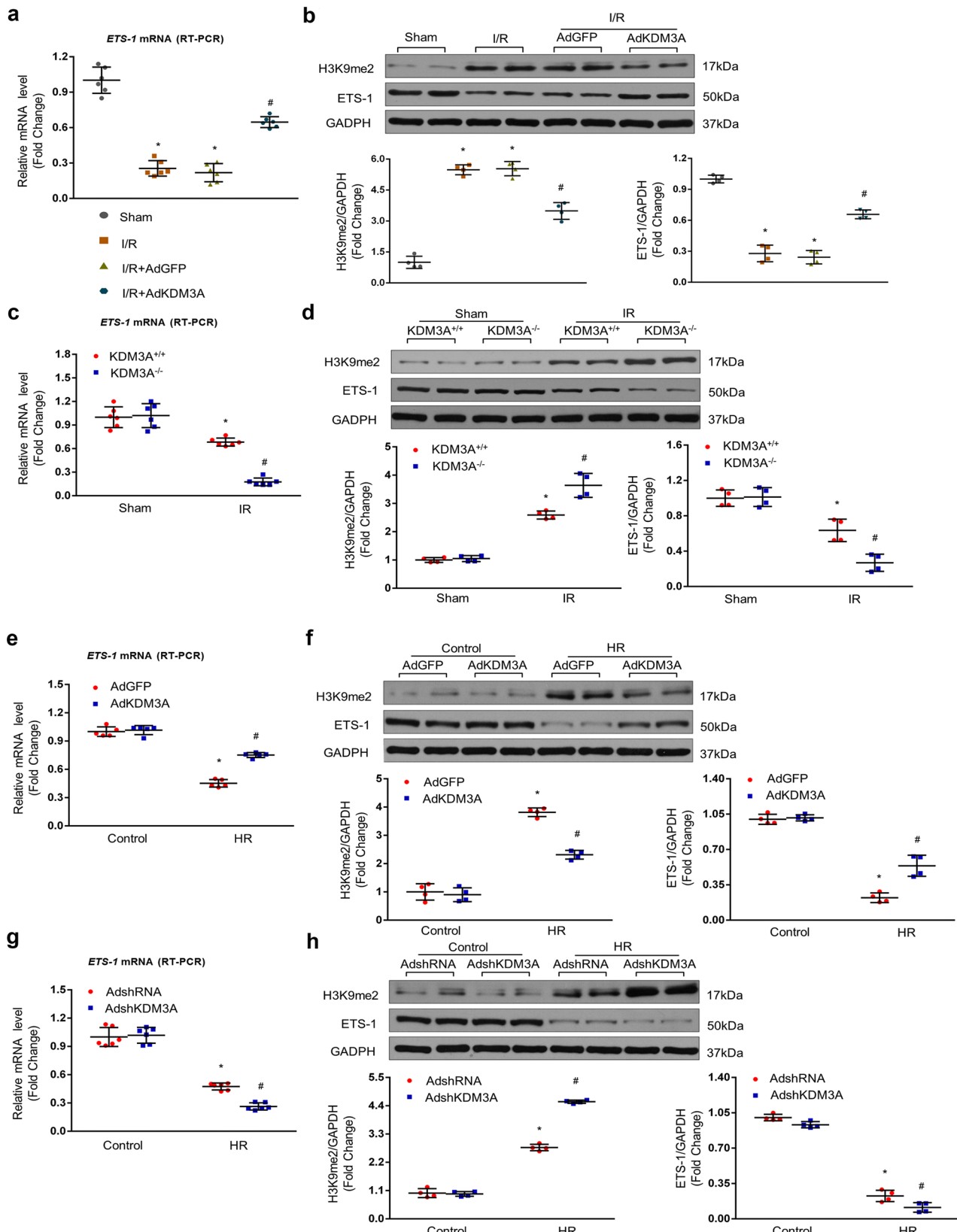

KDM3A acts as an upstream modulator of the MAKP/NF-κB pathways in high insulin-induced vascular smooth muscle cell malfunctions primarily by disrupting inflammation, apoptosis, and ROS. Therefore, this investigation elucidated KDM3A's biological participation in vascular injury with the connections to canonical signaling pathways and the innovative targets.

However, the exact impacts and potential mechanisms of KDM3A in cardiac insult, especially for MIRI are less exploited. Recently, Mathias and coworkers revealed that KDM3A is upregulated in maladaptive cardiac remodeling, and KDM3A correlates positively with increased ANP and BNP expression and inversely with H3K9me2 in the promoter region of ANP[26].

**Fig. 7 KDM3A is involved in MIRI by regulating ETS1 expression in vivo and in vitro. a** Myocardial mRNA expression of the ETS1 gene detected by qRT–PCR with or without adenoviral transfection in response to MIRI or sham ($n = 6$, *$P < 0.05$ vs. sham; #$P < 0.05$ vs. I/R or I/R + AdGFP). **b** Representative western blots (upper panel) and quantitative results (lower panel) showing the protein expression of ETS1 and histone mark H3K9me2 with or without adenoviral transfection undergoing sham or MIRI ($n = 4$, *$P < 0.05$ vs. sham; #$P < 0.05$ vs. I/R or I/R + AdGFP). **c** mRNA levels of the ETS1 gene, as indicated by RT–PCR, in $KDM3A^{+/+}$ and $KDM3A^{-/-}$ rats after sham or MIRI ($n = 6$, *$P < 0.05$ vs. $KDM3A^{-/-}$/sham; #$p < 0.05$ vs. $KDM3A^{+/+}$/IR). **d** Typical western blots (upper panel) and averaged data (lower panel) of ETS1 and H3K9me2 in $KDM3A^{+/+}$ and $KDM3A^{-/-}$ rats with or without MIRI ($n = 4$, *$p < 0.05$ vs. $KDM3A^{-/-}$/sham; #$p < 0.05$ vs. $KDM3A^{+/+}$/IR). **e** mRNA levels of the ETS1 gene detected by qRT–PCR in NRCMs infected with AdKDM3A in response to control or H/R injury ($n = 6$, *$P < 0.05$ vs. AdGFP/Control; #$P < 0.05$ vs. AdGFP/H/R). **f** Representative western blots (upper panel) and quantitative data (lower panel) of the protein levels of ETS1 and H3K9me2 in NRCMs infected with AdKDM3A undergoing control or H/R ($n = 4$, *$P < 0.05$ vs. AdGFP/Control; #$P < 0.05$ vs. AdGFP/H/R). **g** mRNA levels of the ETS1 gene, as indicated by RT–PCR, in NRCMs infected with AdshKDM3A in the absence or presence of H/R injury ($n = 6$, *$P < 0.05$ vs. AdshRNA/Control; #$P < 0.05$ vs. AdshRNA/H/R). **h** Typical western blots (upper panel) and averaged data (lower panel) of ETS1 and H3K9me2 in NRCMs infected with AdshKDM3A after sham or H/R ($n = 4$, *$P < 0.05$ vs. AdshRNA/Control; #$P < 0.05$ vs. AdshRNA/H/R).

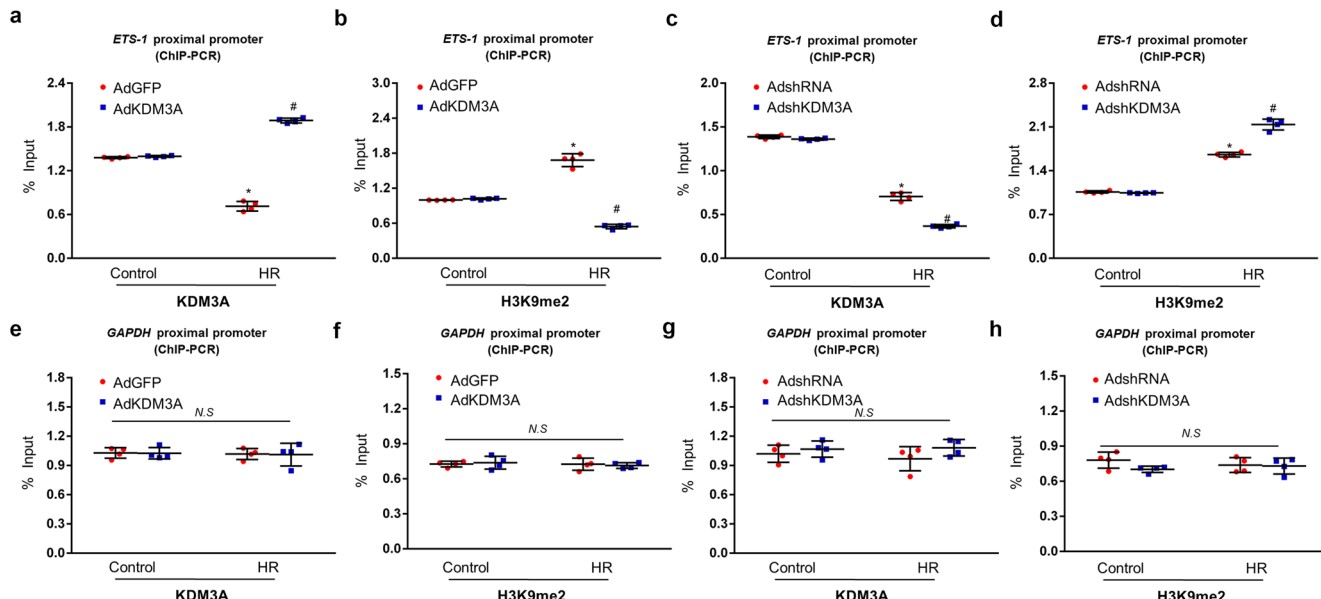

**Fig. 8 ChIP-PCR assay demonstrated that KDM3A regulates the expression of ETS1 by modulating levels of H3K9me2 in its promoter region in the setting of H/R injury. a, b** Primary NRCMs were transfected with AdKDM3A or AdGFP followed by H/R or control. ChIP followed by quantitative PCR (qPCR) with KDM3A and H3K9me2 antibodies. Data are expressed as the fold change compared to the corresponding control group ($n = 4$, *$P < 0.05$ vs. AdGFP/Control; #$P < 0.05$ vs. AdGFP/H/R). **c, d** NRCMs were infected with AdshKDM3A or AdshRNA with or without H/R. ChIP followed by quantitative PCR (qPCR) with KDM3A and H3K9me2 antibodies. Data are expressed as the fold change relative to the corresponding control group ($n = 4$, *$P < 0.05$ vs. AdshRNA/Control; #$P < 0.05$ vs. AdshRNA/H/R). **e–h** The enrichment of KDM3A and H3K9me2 on the GAPDH promoter were regarded as the control ($n = 4$). Data are expressed as the fold change relative to the corresponding control group ($n = 4$, NS: no significant difference).

Connecting these data with current findings, such cardiac-influenced consequences further expand the biochemical actions of KDM3A across cardiovascular disease, albeit with divergent results between pathological cardiac hypertrophy and MIRI.

In summary, our findings raise hope for prospective approaches against MIRI through KDM3A-dependent ETS1 enhancement by impacting multipathogenetic episodes. However, further explorations are still warranted to highlight worthwhile targets directly or indirectly regulated by KDM3A and other types of posttranslational modifications cooperatively. For instance, considering that KDM3A has been verified as a potential target of miRNA-22[44], it was of appealing interest to exploit their epigenetic crosstalk in regulating MIRI based on the existing I/R-regulatory roles of miRNA-22, which seems to permit unrecognized patterns in the prevention of I/R injury.

## Materials and methods

**Adenovirus and reagents**. Adenovirus encoding KDM3A-shRNA (AdshKDM3A) or KDM3A (AdKDM3A) was provided by Genechem (Shanghai, China). AdshRNA or AdGFP was used as a negative control. These viruses were used in our previous studies and were verified to be efficient for the induction of KDM3A knockdown or overexpression both in vivo and in vitro[16,19]. Small interfering RNA (siRNA) against ETS1 (siETS1) and negative control siRNA (siCtrl) were designed and synthesized by RiboBio (Guangzhou, China). Two kinds of KDM3A primary antibodies were purchased from Abcam (Cambridge, UK) (ab106456) for western blotting assay and Santa Cruz (CA, USA) (sc-376608X) for ChIP-PCR detection, respectively. Primary antibodies used in the experiments were purchased from the following suppliers: H3K9me2 (ab1220), Bax (ab32503), Bcl-2 (ab196495), cleaved caspase-9 (ab52298), cleaved caspase-3 (ab2302), cytoplasmic cytochrome $c$ (ab13575), mitochondrial cytochrome $c$ (ab13575), VDAC1 (ab15895) and GAPDH (ab37168) were purchased from Abcam; ETS1 was purchased from Proteintech Group, Inc (12118-1 AP). The secondary antibodies, either HRP-linked goat anti-mouse IgG or goat anti-rabbit IgG, were obtained from BIOSS (Beijing, China). Foetal bovine serum (FBS) was purchased from Hyclone (Waltham, MA, USA). Cell culture reagents and other chemicals were purchased from Sigma (St. Louis, MO. USA) unless otherwise specified. Enhanced chemiluminescence reagents and apoptosis detection kits were obtained from BD Biotechnology. DHE reagent was obtained from Beyotime Biotechnology. Cell Counting Kit-8 (CCK-8) was purchased from Dojindo Molecular Technologies, Inc.

**Animals**. All Sprague-Dawley (SD) rats were bred in a standard environment with controlled temperature (20-25 °C), humidity (40–60%), light conditions (12 h light/dark cycle), and ad libitum access to food and water. For genetic deletion of KDM3A, CRISPR/Cas9 genome-editing technology[45] was used to generate KDM3A-knockout (KDM3A-KO) (hereafter referred to as $KDM3A^{-/-}$) rats (SD

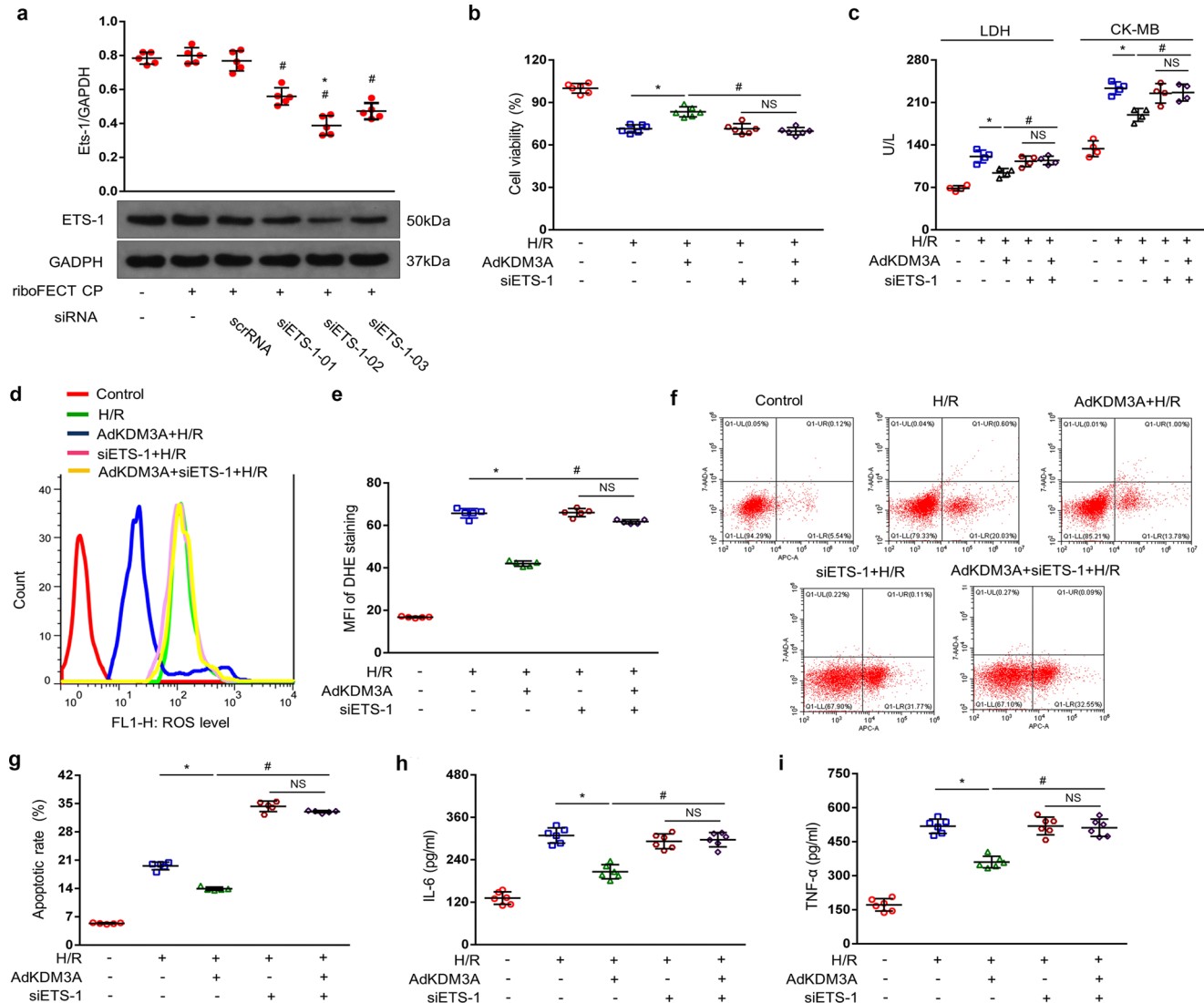

**Fig. 9 KDM3A ameliorates MIRI in an ETS1-dependent manner. a** Representative western blots (upper panel) and quantitative data (lower panel) of the protein levels of three different ETS1 proteins in NRCMs ($n = 5$). *#$P < 0.05$ vs. vehicle. Cell viability was analysed using MTT reagent (**b** $n = 6$) and LDH/ CK-MB concentration (**c** $n = 4$) in NRCMs infected with AdKDM3A followed by siETS1 transfection in response to H/R or control. ROS accumulation was detected by DHE staining and flow cytometry (**d** left panel), and the MFI (**e** right panel) was quantified in NRCMs ($n = 5$). Apoptosis was detected by flow cytometry using 7-AAD/AV-APC staining (**f**) and the apoptotic rate was quantified (**g**) ($n = 5$). **h**, **i** Quantitative results of IL-6/TNF-α in the culture medium of cardiomyocytes ($n = 6$). *$P < 0.05$ vs. H/R; #$P < 0.05$ vs. H/R/AdKDM3A.

background). SD rats were obtained from the Vital River (Strain code: 101, Beijing, China). One single guide RNA (sgRNA) flanked exon 5 of *KDM3A* gene in rat was designed and created. When purified Cas9 mRNA and sgRNA were mixed and microinjected into embryos, PCR products were TA cloned and sequenced to define the exacted indel mutations of generated founders. The primers were listed as follows: *KDM3A* F: 5'-TCCCTGGAGTTGCAGTAGTTT-3'; *KDM3A* R: 5'-AGGTCTATCACTGGCTAACTCA-3'. Out of the overall 29 embryos, 8 pups were generated in which three founders were measured to be multiple mutated allele (#70-13; #70-10; #71-1) and identified as mosaics. Founder #70-13 was chosen as the breeding line and mated with the WT *(KDM3A+/+)* SD rat strain to generate F1 heterozygote *(KDM3A +/−)*. The homozygous *KDM3A −/−* strain was obtained through sibling mating of heterozygous F1 offspring and was identified by sequencing chromatograms of PCR products. The western blotting assay was performed to validate KDM3A protein expression. To induce myocardial over-expression of KDM3A in vivo, male SD rats (SPF grade, weighing 220–250 g) were purchased from Vital River Laboratories (VRL, Beijing, China). Cardiac-specific KDM3A overexpression in vivo was garnered by intramyocardial injection of AdKDM3A[32,33]. Male SD rats (SPF grade, weighing 220–250 g) were purchased from Vital River Laboratories (VRL, Beijing, China). The intramyocardial injection of 100 ul solution of AdKDM3A ($1 \times 10^{10}$ pfu/ml) were performed at five separate sites as depicted in our previous description. Three days later, the MIRI model was operated as following delineations and as our prior experiments. The experiments and all animal care outlined in our study were performed in adherence with the

Guide for the Care and Use of Laboratory Animals published by the US National Institutes of Health (NIH Publication, 8th Edition, 2011) and were approved by the Institutional Animal Care and Use Committee of Wuhan University.

**MIRI model.** The myocardial I/R model was generated[9,34]. In brief, rats were anaesthetized by an intraperitoneal injection of pentobarbital sodium (40 mg/kg) (Sigma). Adequate anaesthesia was confirmed by the loss of a toe pinch reflex. A left parasternal incision was performed after rats were anaesthetized. The left anterior descending artery was then ligated for 30 min, and 24 h reperfusion was subsequently performed. The chest cavity, muscle and skin were closed in layers. Myocardial tissues from the infarct area (IA) and area at risk (AAR) (5 mm around the IA) were collected to ensure detection after I/R procedures. Sham-operated rats underwent the same procedures without left anterior descending artery occlusion or reperfusion.

**Immunofluorescence assay.** Expression of KDM3A in rat hearts was observed by immunofluorescence[5,32]. In brief, paraffin-embedded sections were incubated in PBS containing 10% goat serum for 1 h and consecutively incubated with primary antibodies against KDM3A (Abcam, ab106456) and α-actinin (Millipore, 05-384). Next, the sections were incubated with appropriate secondary antibodies, including the FITC-conjugated goat anti-rabbit IgG (Boster, BA1105) and Cy3-conjugated goat anti-mouse IgG (Boster, BA1031). The 4',6-Diamidino-2-phenylindole (DAPI,

Sigma) was used to stain the cell nuclei. Images were acquired using a fluorescence microscope (OLYMPUS DX51, Tokyo, Japan) in at least 5 random fields per sample. Staining per section was estimated in a blinded method (400× magnification). In addition, the same methods were performed in primary cardiomyocytes cultured on 6-well chamber slides as in previous demonstrations.

**Echocardiographic and Evans blue+TTC estimations**. Echo assay was performed to determine cardiac left ventricular (LV) functions after 24 h after reperfusion[22,46]. LV M-mode tracing at mid-papillary levels was measured and averaged for the left ventricular ejection fraction (LVEF) and fractional shortening (LVFS).

Twenty-four hours following I/R injury, the left anterior descending artery was religated at the same site of prior occlusion, and myocardial IA was determined[32] by Evans blue and TTC staining. Each section was imaged using a microscope camera and quantified using Image-Pro Plus 6.0 (Media Cybernetics, Bethesda, MD, USA). The percentage of myocardial infarction was calculated using the following formula: IA/total left ventricles × 100%.

**Terminal uridine nick-end labeling (TUNEL)**. Paraffin sections were stained using an in situ TUNEL detection kit (Roche, Basel, Switzerland) to identify apoptotic cells[7]. Anti-α-actinin antibody, red fluorescent dye, and DAPI were used to specifically identify cardiomyocytes, apoptotic nuclei, and total nuclei, respectively. TUNEL-positive nuclei (red) and total cells (blue) were imaged using an Olympus DX51 fluorescent microscope (Olympus, Tokyo, Japan) in at least five randomly selected fields per section (400× magnification). The apoptotic index is expressed as the ratio of apoptotic cells to total myocytes of DAPI-stained nuclei.

**Mitochondrial morphology**. Ventricular specimens were collected from the apex territory to the left anterior descending artery occlusion region at 24 h after reperfusion. The dissected heart tissue in 1–2 mm wide blocks was immersed in 4% glutaraldehyde overnight, After the samples were postfixed in 1% osmium tetroxide for 1 h, the sections were dehydrated in a graded ethanol series to 100% and embedded in epoxy resin. Ultrathin sections of 80 nm thick were observed using a JEM-1400 transmission electron microscope (TEM) (JEOL, TOKYO, JAPAN). Mitochondria were imaged at high magnification (×6000)[36].

**Cell culture and treatment**. Neonatal rat cardiomyocytes (NRCMs) were isolated from the hearts of SD rats at 1–3 d after birth and cultured in DEEM/F12 medium[9,34,46]. In brief, hearts from anesthetized rats were collected, intersected into pieces and digested with 0.125% trypsin and 0.08% collagenase II. After cells were pre-plated for 90 min, nonadherent cells were collected and re-seeded into 6-well plates at a density of $1 \times 10^6$ cells/well. Then, the cells were cultured in complete DMEM/F12 medium consisted of 10% FBS and 1% penicillin/streptomycin for 48 h before the following treatments. For adenoviral infection, primary NRCMs were infected with Ad-KDM3A/Ad-GFP at a multiplicity of infection (MOI) of 50 or Ad-shKDM3A/Ad-shRNA at an MOI of 50 for 4 h. siETS1 or siCtrl (as a control) transfection was performed in line with the instructions suggested by the manufacturer. For experimental H/R injury, isolated primary cardiomyocytes were seeded into culture plates, and fresh medium was added prior to each procedures. An H/R model was established as previously demonstrated. Briefly, hypoxia was induced in an anaerobic incubator chamber (Thermo Forma, Thermo Fisher Scientific, Boston, MA, USA) suffused with a gas mixture composed of 95% N2 and 5% CO2 at 37 °C, and cells were incubated with serum- and glucose-deprived medium for 4 h. To mimic reperfusion, the medium was replaced with 10% DMEM/F12 medium and cells were removed to normoxic condition containing 95% O2 and 5% CO2 for 24 h to establish reoxygenation. Meanwhile, control cells were maintained in normoxic incubator for equivalent durations.

**Measurements of LDH/CK-MB and IL-6/TNF-α**. To assay necrotic enzymes released by cardiomyocytes, blood samples and cellular supernatants were collected for the biological analysis of LDH and CK-MB using commercial analytical kits[32,47] (Beijing Kemeidongya Biotechnology Ltd, China). The results were determined in international units per litre. The levels of proinflammatory mediators such as IL-6/TNF-α in cardiomyocytes were detected using commercial ELISA kits (Nanjing Jiancheng Bioengineering Institute, Nanjing, China) according to the manufacturer's instructions.

**Assessment of ATP content**. ATP content was detected using a bioluminescent assay kit (Beyotime, Nanjing, China) according to the instruction[36]. In brief, cells were homogenized in lysis buffer and centrifuged at 12,000 g for 10 min at 4 °C. The collected supernatant was mixed with assessment reagent, and a microplate reader was used to measure ATP content on a luminescence luminometer. The levels of ATP in heart tissues were measured in the same way.

**Cell viability, apoptosis and ROS assays**. Isolated primary cardiomyocytes were seeded into 96-well plates, and cell viability was assessed using 3-(4,5-dimethyl-thiazol-2-yl)-2,5-diphenyltertrazolium bromide (MTT) reagent (Solarbio, China). Cell viability was expressed as a percentage of the controls. Flow cytometry was

used to estimate apoptotic cells. The cells underwent staining with AV-APC and 7-AAD dyes, followed by fluorescence-activated cell sorting on a flow cytometric assay (CytoFLEX; Beckman Coulter, Inc., Brea, CA, USA). Apoptotic cells were defined as those that were AV-APC-positive and 7-AAD-negative cells. To determine the levels of ROS ($O_2^-$ in particular) in NRCMs, cells were labeled with a dihydroethidium (DHE) probe (Beyotime Institute of Biotechnology Co., Ltd., Haimen, China) and subjected to flow cytometry assay (FACSCalibur; BD Biosciences, Franklin Lakes, NJ, USA)[48]. To further detect intercellular superoxide generation and lipid peroxidation induced by H/R injury, the MDA concentration and SOD enzymatic activity in NRCMs were measured using the same methods in cardiac tissue, in accordance with the descriptions of commercial kits (Nanjing Jiancheng Bioengineering Institute, China)[48]. To evaluate tissue generation of ROS, left ventricular samples (5-μm sections) were probed with DHE (2 mmol/L) for 1 h at room temperature in the dark as previously introduced[41]. The slides were visualized by fluorescence microscopy (Olympus DX51, Tokyo, Japan), and the mean fluorescence intensity (MFI) of each slide was quantified by Image-Pro Plus software version 6.0.

**Cytosol and mitochondrial protein extraction**. A mitochondrial protein extraction kit (EnoGene, Nanjing, China) was utilized to obtain cytosolic and mitochondrial fractions. Consistent with the manufacturer's instructions, cells or fresh tissue were collected, homogenized in ice-cold lysis buffer I and centrifuged for 5 min at 1000 g. Then, the supernatant was added to 500 μl of mitochondrial isolation reagent followed by centrifugation at 15,000 g for 20 min. The supernatant containing the cytosolic fraction was harvested, and the precipitate was washed with rinsing buffer. After centrifugation at 15,000 g for 10 min, the supernatant was discarded, and ice-cold lysis buffer II was chosen to resuspend the pellet. After 15 min of centrifugation (12,000 rpm), the supernatant (containing mitochondrial proteins) was obtained. Protein levels of cytochrome c in the cytosol and mitochondria were detected by western blotting assay.

**Chromatin immunoprecipitation (ChIP)-PCR**. ChIP was performed, and ChIP-enriched DNA was analysed by real-time PCR[4,9,19]. In brief, treated cardiomyocytes were fixed and cross-liked with 1% formaldehyde for 20 min at room temperature. After terminating cross-linking by 0.125 M glycine for 5 min, cell lysates were sonicated into chromatin fragments of approximately 400 bp in length. The samples were subjected to immunoprecipitation using 5 μg of antibody against H3K9me2 (Abcam) or nonspecific IgG control (Santa Cruz) in the presence of magnetic beads conjugated with secondary antibody. Immune complexes and input were washed and eluted with buffer. Then, ChIP-enriched DNA fragments were assayed by real-time PCR using primers close to the promoter sites on the ETS1 with GAPDH used as a control. The primer sequences were listed as follows: ETS1, (forward, 5′-AGGGTGGAGATGGGAGATGTGA-3′; reverse, 5′-AGGGTGGA-GATGGGAGATGTGA-3′); GAPDH, (forward, 5′-GCCGAAGTACCCAAGGA-GACC-3′; reverse, 5′-AGCAAAGGCGGAGTTACAAGG-3′).

**Quantitative RT-PCR and Western blotting analyses**. Quantitative RT-PCR (qRT-PCR) were was performed as previously demonstrated[4,9,19]. In brief, total RNA from cardiac tissue was extracted using TRIzol reagent (Invitrogen) according to the manufacturer's instructions. Two micrograms of mRNA were reverse-transcribed into cDNA using a cDNA synthesis kit (Fermentas). RT-PCR was performed using a SYBR green/fluorescein qPCR Master Mix kit (Fermentas) with an ABI Prism 7500 system. Data were normalized to β-actin to indicate relative expression levels. The sequence-specific primers used are listed as follows:

KDM3A, F: 5′-TGAGGGCCTCTGTGAAATGT-3′, R: 5′- GGAAGCACTGAT TTGGCACA -3′;
ETS-1, F: 5′- CCCAGAATCCCGTTACACCT-3′, R: 5′- GTGTCTGTCTGGA GAGGGTC -3′.

For western blotting assays, frozen heart tissue was lysed[4,9,19]. After the loading concentration was essentially normalized to 50 μg per well, extracted proteins were separated by 8–12% SDS-PAGE and then transferred onto PVDF membranes (Merck Millipore). Next, the membranes were blocked in 5% skim milk powder dissolved in Tris-buffered saline containing 0.1% Tween-20, washed three times and subsequently probed with the corresponding primary antibodies at the recommended dilution overnight at 4 °C. After incubation with horseradish peroxidase-conjugated secondary antibodies for 60 min, the bands visualized using enhanced chemiluminescence reagent. GAPDH was used as a loading control for whole cell lysate and cytosolic protein, and VDAC1 was used as a loading control for the targets in mitochondria.

**Statistics and reproducibility**. The numerical data were represented as the mean ± SEM. Statistical p values between two groups were calculated using the Student's t-test. Comparisons of more than two groups were assessed by one-way analysis of variance (ANOVA) and a post hoc Tukey's test. Statistical significance was assigned at a p value less than 0.05. All results were independently reproduced at least three times with similar results.

**Reporting summary**. Further information on research design is available in the Nature Research Reporting Summary linked to this article.

## Data availability
The source data for the graphs and charts in the main figures as well as the original uncropped blot/gel images are provided in Supplementary information. All other data are available from the corresponding author on reasonable request.

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

## Acknowledgements

This work was supported by National Natural Science Foundation of China (81570331, 81770360, 81900334, 82070372), Hubei Province's Outstanding Medical Academic Leader Program, Hubei Province Health and Family Planning Scientific Research Project (WJ2019Z004).

## Author contributions

X.G., B.Z., and J.Z.performed the study, analyzed, and interpreted the data, and wrote the manuscript. B.Z., J.C., G.L., and Q.H. made substantial contributions to the acquisition of data and manuscript revision. J.C. and X.G.performed experimental hypotheses, participated in the experimental design and data revision. X.G., J.C., B.Z., and J.Z.performed the interpretation of data and manuscript drafting. All authors read and approved the final manuscript.

## Competing interests

The authors declare no competing interests.
