## [Peer Review File · Communications Biology]

Reviewers' comments:

Reviewer #1 (Remarks to the Author):

In this manuscript authors Xin Guo et al show that KDM3A is a protective factor in I/R injury model. KDM3A protein is decreased in both in vitro and in vivo I/R model. Authors show that I/R injury phenotype is ameliorated when KDM3A is overexpressed, while in the absence of KDM3A, I/R injury is more severe. Mechanically, KDM3A regulates I/R injury responses through regulating ETS1 expression. The manuscript is well written.

Major concerns:

1. In fig6, Western blot results show H3k9Me2 levels are unchanged no matter when KDM3A is knocked down or over expressed at basal level. Authors need to explain this phenomenon.
2. In fig7, while ETS1 is show to mediate KDM3A's role in regulating apoptosis, ROS and inflammation, is there any proof showing ETS1 have this function in any animal model?

Minor concerns:

1. Line 256, what "Inconsistent" means here?
2. Figure legend of Fig6 saying "KDM3A-mediated cardioprotection required the activation of ETS1 in MIRI", but what the figures show is the level of ETS1.
3. Authors name "Jing Zhang" becomes "Zhang Jing" in "Author contribution" section.

Reviewer #2 (Remarks to the Author):

This study identified KDM3A as a protective regulator against MIRI both in vivo and in vitro by utilizing gain- and loss-of-functional approaches. Then it devoted to the potential molecular mechanisms. After a series of experiments, the authors indicated that KDM3A elicits protective contributions in apoptosis, ROS and inflammation-associated processes, largely dependent on the activation of ETS1 during MIRI. Thus, this study proposed that KDM3A emerges as a potent therapeutic candidate for the prevention of MIRI.

The work of this paper is rigorous and logical. The authors have done a lot of work in this field. The result of the study is significant for the treatment of myocardial I-R injury.

I would suggest the following minor comment:

- 1) As a large number of experiments were carried out in this study, many graphs were presented in each part of the paper. It would be better to optimize the number of graphs in each figure or the layout of the graphs.
- 2)As the authors issued in the study limitation, further explorations are still warranted to highlight worthwhile targets directly or indirectly regulated by KDM3A and other types of PTMs cooperatively.

Reviewer #3 (Remarks to the Author):

The work of Xin Guo et al elegantly provides evidence that KDM3A protects the myocardium from ischemia/reperfusion injury via the modulation of the ETS1 transcription factor. The authors explored KDM3A's protective function in the heart by investigating in vitro and in vivo models.

I like this work and do not have major concerns.

I invite the authors, however, to revise carefully the manuscript's language before resubmitting. In addition, the quality of the fluorescence panels with red dyes should be improved. It is currently barely visible and certainly below standard.

Responses to the comments of Reviewers

Reviewer: 1

Major concerns:

1. In fig6, Western blot results show H3k9Me2 levels are unchanged no matter when KDM3A is knocked down or over expressed at basal level. Authors need to explain this phenomenon.

Response: This is a very interesting phenomenon, as shown in our present study, the levels of H3K9me2 and the expression of ETS1 were unchanged at the basal level no matter when KDM3A knockdown or overexpression. Besides, results of the subsequent ChIP-PCR assay also indicated that the enrichments of KDM3A and H3K9me2 on the promoter region were also unchanged at the basal level. likewise, results from another study published by our research group aiming to explore the potential function of KDM3A in modulating macrophage polarization and alleviating post-MI ventricular remodeling also uncovered that the levels of H3K9me2 were unchanged after KDM3A knockout at the basal level [1]. Moreover, similar results have been found in studies by other researchers as well. For example, as a histone H3K9 trimethyltransferase, SUV39H1 deficiency or inhibition attenuated MIRI-induced infarction and improved heart function in mice through influencing ROS levels in a SIRT1-dependent manner. In this paper, researchers also find SUV39H1 knockout has no obvious effect on the levels of SIRT1 and H3K9me3 as

well as the expression of SIRT1 downstream target gene SOD1 and SOD2 at the basal level [2].

As we know, the methylation level of histone is regulated by both histone demethylase and histone methyltransferase, and is in dynamic equilibrium the physiological state. The activities of histone demethylases (including LSD1 and the JMJC domain-containing proteins) and histone methyltransferases (including SUV39H1, G9a, and others) are to remove or deposit methyl group at lysine 9 of H3 (H3K9) from pathogenic gene promoter respectively. In fact, a lot of histone demethylases and histone methyltransferases were involved in the modulation of H3K9me2 (Table 1), KDM3A is just one of the histone demethylases that regulate H3K9me2 [3]. Therefore, we speculate that regulation of a single demethylase or methyltransferase may not be sufficient to alter this equilibrium and the degree of methylation at the basal level. However, in certain diseases such as MIRI and myocardial infarction, histone demethylase KDM3A and histone methyltransferase SUV39H1 may play a central role in the regulatory network of H3K9me2. Thus, they were able to regulate disease progression by altering H3K9me2 levels in the promoter region of key downstream target molecules. More experiments will be conducted to confirm this hypothesis.

[1]Xiaopei, Liu, Jing, Chen, Bofang, & Zhang, et al. Kdm3a inhibition modulates macrophage polarization to aggravate post-mi injuries and accelerates adverse ventricular remodeling via an irf4 signaling pathway - sciencedirect. Cellular

Signalling, 64, 109415-109415.

[2] Yang G, Zhang X, Weng X, Liang P, Dai X, Zeng S, Xu H, Huan H, Fang M, Li Y, Xu D, Xu Y. SUV39H1 mediated SIRT1 trans-repression contributes to cardiac ischemia-reperfusion injury. *Basic Res Cardiol.* 2017 May;112(3):22. doi: 10.1007/s00395-017-0608-3. Epub 2017 Mar 8. PMID: 28271186.

[3] Black JC, Van Rechem C, Whetstone JR. Histone lysine methylation dynamics: establishment, regulation, and biological impact. *Mol Cell.* 2012 Nov 30;48(4):491-507. doi: 10.1016/j.molcel.2012.11.006. PMID: 23200123; PMCID: PMC3861058.

Histone demethylases	Histone methyltransferases
LSD1/KDM1A	PRDM3/KMT8E
AOX1/KDM1B	PRDM8/KMT8D
JMJD1A/KDM3A	PRDM16/KMT8F
JMJD1B/KDM3B	G9a/EHMT2
JMJD1C/KDM3C	EHMT1/KMT1D
PHF8/KDM7B	SUV39H1/KMT1A
JMJD2A/KDM4A	SUV39H2/KMT1B
JMJD2B/KDM4B	ESET
JMJD2C/KDM4C	CLLD8/KMT1F
JMJD2D/KDM4D	PRDM2/KMT8A
KDM4E	

Table 1, Histone demethylases and histone methyltransferases involved in the modulation of H3K9me2.

2. In fig7, while ETS1 is show to mediate KDM3A's role in regulating apoptosis, ROS and inflammation, is there any proof showing ETS1 have this function in any animal model?

Response: Thank you very much for your valuable recommendation. We reviewed the latest and previously published works of literature again. Numerous studies have varied ETS1 is a key downstream target of KDM3A. For example, Lays M et al., recently provide direct clues that KDM3A/ETS1 epigenetic axis plays an important role in disease promotion in Rhabdomyosarcoma (RMS) by facilitating RMS cells proliferation, invasion, and survival [1]. In a nude mice cervical cancer model, Liu and his colleagues indicated that KDM3A mediated ETS1 activity through the demethylation and histone modification of H3K9me2, thus inhibiting the expression of apoptosis-associated proteins in cervical epithelial cells [2]. Also, our research team's recently published work demonstrated that KDM3A could ameliorate acute myocardial infarction (AMI)-induced cardiomyocytes apoptosis and alleviate inflammatory-response by regulating the expression of EST1[3].

For EST1, as mentioned in the discussion section of our present manuscript, in a rat's acute renal failure model, ETS1 stimulation is responsible for apoptotic limitation in renal ischemia and reperfusion injury [4]. Besides, in an Ang II-caused vascular remodeling mice model, Ni et al., uncovered that ETS1 knockout could

diminish ROS generation and inflammation [5]. Moreover, Bian and his colleagues also provided evidence that ETS1 overexpression could reduce mitochondrial apoptosis in a rat myocardial ischemia/reperfusion injury [6].

In our present study, our data indicated that KDM3A exerts an anti-apoptotic, anti-ROS, and anti-inflammation efficiency by regulating EST1 in the context of myocardial ischemia/reperfusion injury.

[1] Sobral LM, Hicks HM, Parrish JK, McCann TS, Hsieh J, Goodspeed A, Costello JC, Black JC, Jedlicka P. KDM3A/Ets1 epigenetic axis contributes to PAX3/FOXO1-driven and independent disease-promoting gene expression in fusion-positive Rhabdomyosarcoma. *Mol Oncol.* 2020 Oct;14(10):2471-2486. doi: 10.1002/1878-0261.12769. Epub 2020 Aug 5. PMID: 32697014.

[2] Liu J, Li D, Zhang X, Li Y, Ou J. Histone Demethylase KDM3A Promotes Cervical Cancer Malignancy Through the ETS1/KIF14/Hedgehog Axis. *Onco Targets Ther.* 2020 Nov 19;13:11957-11973. doi: 10.2147/OTT.S276559. PMID: 33239895.

[3] Zhang BF, Jiang H, Chen J, Hu Q, Yang S, Liu XP, Liu G. LncRNA H19 ameliorates myocardial infarction-induced myocardial injury and maladaptive cardiac remodeling by regulating KDM3A. *J Cell Mol Med.* 2020 Jan;24(1):1099-1115. doi: 10.1111/jcmm.14846. Epub 2019 Nov 21. PMID: 31755219.

[4] Tanaka H, Terada Y, Kobayashi T, Okado T, Inoshita S, Kuwahara M, Seth A, Sato Y, Sasaki S. Expression and function of Ets-1 during experimental acute renal failure in rats. *J Am Soc Nephrol.* 2004 Dec;15(12):3083-92. doi:

10.1097/01.ASN.0000145459.54236.D3. PMID: 15579511.

[5] Ni W, Zhan Y, He H, Maynard E, Balschi JA, Oettgen P. Ets-1 is a critical transcriptional regulator of reactive oxygen species and p47(phox) gene expression in response to angiotensin II. *Circ Res.* 2007 Nov 9;101(10):985-94. doi: 10.1161/CIRCRESAHA.107.152439. Epub 2007 Sep 13. PMID: 17872466.

[6] Bian C, Xu T, Zhu H, Pan D, Liu Y, Luo Y, Wu P, Li D. Luteolin Inhibits Ischemia/Reperfusion-Induced Myocardial Injury in Rats via Downregulation of microRNA-208b-3p. *PLoS One.* 2015 Dec 14;10(12):e0144877. doi: 10.1371/journal.pone.0144877. PMID: 26658785.

Minor concerns:

1. Line 256, what "Inconsistent" means here?

Response: Thank you for your valuable comment. What we want to point out here is that KDM3A overexpression could significantly reduce myocardial infarct size, improve cardiac function as well as reduce the levels of LDH and CK-MB in the setting of I/R injury. We are deeply sorry for the mistake. In the revised manuscript, the word “Inconsistent” has been revised by “In consistent with these phenomena”.

2. Figure legend of Fig6 saying "KDM3A-mediated cardioprotection required the activation of ETS1 in MIRI", but what the figures show is the level of ETS1.

Response: Maybe we didn't make it clear, here we want to elaborate that ETS1 is a key downstream target of KDM3A in the context of MIRI and KDM3A regulates

MIRI by regulating ETS1 expression. We measured the expression of ETS1 in mRNA and protein levels both in vivo and in vitro after upregulating or downregulating the expression of KDM3A in the setting of I/R injury. As shown in the original figure 6, figure 6A and figure 6B (revised figure7A and figure 7B) show that myocardial mRNA and protein expression of the ETS1 were significantly upregulated in the setting of I/R injury after KDM3A was upregulated in vivo. However, figure 6C and figure 6D (revised figure7C and figure 7D) show that myocardial mRNA and protein expression of the ETS1 were significantly downregulated in the setting of I/R injury after KDM3A was knockout in vivo. Besides, figure 6E and figure 6F (revised figure7E and figure 7F) show that the mRNA and protein expression of the ETS1 were evidently upregulated in the setting of H/R injury after KDM3A was upregulated in vitro. Figure 6G and figure 6H (revised figure7G and figure 7H) show that the mRNA and protein expression of the ETS1 were evidently downregulated in the setting of H/R injury after KDM3A was downregulated in vitro. The following experiments performed by ChIP-PCR assay further determined KDM3A could bind to the promoter region of ETS1 and activate the transcription of ETS1 (original figure6I-L, revised figure7A-D). Moreover, we also performed a rescue experiment to verify KDM3A ameliorates MIRI in an ETS1-dependent manner (original figure 7, revised figure8). In the revised manuscript, the Figure legend “KDM3A-mediated cardioprotection required the activation of ETS1 in MIRI.” was replace by “KDM3A is involved in MIRI by regulating the expression of ETS1 both in vivo and in vitro”.

3. Authors name "Jing Zhang" becomes "Zhang Jing" in "Author contribution" section.

Response: Thank you for your valuable comment. We are deeply sorry for the mistake. In the revised manuscript, the name “Zhang Jing” in "Author contribution" section has been revised by “Jing Zhang”.

Reviewer: 2

I would suggest the following minor comment:

1) As a large number of experiments were carried out in this study, many graphs were presented in each part of the paper. It would be better to optimize the number of graphs in each figure or the layout of the graphs.

Response: Thank you very much for your valuable recommendation. We also noticed there are too many graphs in the original figure 4 and figure 6. According to your advice, we split the contents of the original Figure 4 and Figure 6 into two separate figures. The revised figures are much cleaner and show our work more clearly.

2)As the authors issued in the study limitation, further explorations are still warranted to highlight worthwhile targets directly or indirectly regulated by KDM3A and other types of PTMs cooperatively.

Response: As we mentioned in the study limitation, further experiments are still needed to explore other worthwhile targets directly or indirectly regulated by KDM3A and also to uncover the upstream regulators of KDM3A. Our research group

has devoted ourselves to studying the relationship between KDM3A and cardiovascular disease. We believe KDM3A is a central mediator in the regulatory network of cardiovascular disease. For example, in a rat model of myocardial infarction (MI), our data uncovered that KDM3A deficiency could regulate macrophage polarization to aggravate the inflammatory response and accelerate LV remodeling in vivo. Among them, we confirmed that IRF4 is a downstream effector of the KDM3A-dependent pathway which could epigenetically influence the transcription of IRF4 by enhancing H3K9me2 accumulation on the IRF4 gene proximal promoter region to modulate macrophage polarization. These results demonstrated that KDM3A plays an essential role in the cardiac repair process of post-MI and LV remodeling by modulating the macrophage phenotype, thereby suggesting a promising therapy to treat post-MI injuries [1]. Besides, our recently published paper also verified KDM3A function as a key downstream regulator of the LncRNAH19/miRNA-22-3p signaling pathway and involved in post-translational modifications. Our experiment provides compelling evidence that LncRNA H19 could modulate the expression of KDM3A by competitively binding to miR-22-3p, as a result, the inhibition of miR-22-3p on KDM3A mRNA degradation and transcriptional activity was relieved, which consequently ameliorated acute myocardial infarction-induced myocardial damage and cardiac remodeling [2]. Moreover, other researchers also demonstrated KDM3A takes an active part in left ventricular hypertrophy and fibrosis in response to pressure-overload by methylation regulating the transcription of Timp1[3]. Noteworthy, our ongoing experiments also

attempt to confirm that KDM3A regulates cardiac microcirculation endothelial cells function through PI3K/Akt signaling pathway and participates in myocardial I/R injury. Therefore, along with the deepening of research, more and more targets directly or indirectly regulated by KDM3A and other types of PTMs will find out.

[1]Xiaopei, Liu, Jing, Chen, Bofang, & Zhang, et al. Kdm3a inhibition modulates macrophage polarization to aggravate post-mi injuries and accelerates adverse ventricular remodeling via an irf4 signaling pathway - sciencedirect. Cellular Signalling, 64, 109415-109415.

[2]Zhang BF, Jiang H, Chen J, et al. LncRNA H19 ameliorates myocardial infarction-induced myocardial injury and maladaptive cardiac remodelling by regulating KDM3A. J Cell Mol Med. 2020;24(1):1099-1115.

[3]Zhang QJ, Tran TAT, Wang M, et al. Histone lysine dimethyl-demethylase KDM3A controls pathological cardiac hypertrophy and fibrosis. Nat Commun. 2018 Dec 7;9(1):5230.

Reviewer: 3

I invite the authors, however, to revise carefully the manuscript's language before resubmitting. In addition, the quality of the fluorescence panels with red dyes should be improved. It is currently barely visible and certainly below standard.

Response:

1. We are deeply sorry for the inconvenience due to our negligence. We proofread the

manuscript carefully and tried our best to minimize grammar and spelling errors. Meanwhile, we invited our native English-speaking colleagues to help us revise the manuscript again. Moreover, we have also invited a professional language polishing agency to help us edit and revise our manuscript. We have enclosed the language certificate issued by the agency of AJE. We hope that the revised manuscript will clearly elaborate our experiment and viewpoint.

2. For the quality of the fluorescence panels with red dyes, we think you might be referring to the DHE staining in the original figure 3G (revised figure 3G) and figure 4L (revised figure 5G). The ROS production of myocardial tissue in the sham groups is much less than that in the I/R groups. Therefore, the mean fluorescence intensity (red fluorescence intensity) of the sham group is much weaker than that in the I/R groups. Besides, the resolution of the figures we upload in the submission system is much higher than that in the PDF version, the red fluorescence of ROS and TUNEL are much brighter and clearer in the figures that we upload in the submission system. However, many graphs were presented in our original figures, which might make some details hard to discern. According to the recommendation of Reviewer 2, we optimized the number of graphs in each figure and the layout of the graphs in the revised manuscript. Images become clearer as there are fewer graphs in each figure.

REVIEWERS' COMMENTS:

Reviewer #1 (Remarks to the Author):

I am satisfied with the revision. recommend for publication.

Reviewer #2 (Remarks to the Author):

After modification, I think this paper has reached the publication demand, and my suggestion is that this paper could be accepted. Thanks!

Reviewer #3 (Remarks to the Author):

The authors answered my questions and no further review is necessary.

Response to Referees

REVIEWERS' COMMENTS:

Reviewer #1 (Remarks to the Author):

I am satisfied with the revision. recommend for publication.

Response: Thank the reviewer for reviewing our revised manuscript. We are so happy that you have recommended the acceptance of the manuscript.

Reviewer #2 (Remarks to the Author):

After modification, I think this paper has reached the publication demand, and my suggestion is that this paper could be accepted. Thanks!

Response: Thank the reviewer for reviewing our revised manuscript. We are so happy that you have recommended the acceptance of the manuscript.

Reviewer #3 (Remarks to the Author):

The authors answered my questions and no further review is necessary.

Response: Thank the reviewer for reviewing our revised manuscript.